# Dueling Bandits: Beyond Condorcet Winners to General Tournament Solutions

**Siddartha Ramamohan**
Indian Institute of Science
Bangalore 560012, India
siddartha.yr@csa.iisc.ernet.in

**Arun Rajkumar**
Xerox Research
Bangalore 560103, India
arun_r@csa.iisc.ernet.in

**Shivani Agarwal**
University of Pennsylvania
Philadelphia, PA 19104, USA
ashivani@seas.upenn.edu

## Abstract

Recent work on deriving $O(\log T)$ anytime regret bounds for stochastic dueling bandit problems has considered mostly Condorcet winners, which do not always exist, and more recently, winners defined by the Copeland set, which do always exist. In this work, we consider a broad notion of winners defined by tournament solutions in social choice theory, which include the Copeland set as a special case but also include several other notions of winners such as the top cycle, uncovered set, and Banks set, and which, like the Copeland set, always exist. We develop a family of UCB-style dueling bandit algorithms for such general tournament solutions, and show $O(\log T)$ anytime regret bounds for them. Experiments confirm the ability of our algorithms to achieve low regret relative to the target winning set of interest.

## 1 Introduction

There has been significant interest and progress in recent years in developing algorithms for dueling bandit problems [1–11]. Here there are $K$ arms; on each trial $t$, one selects a pair of arms $(i_t, j_t)$ for comparison, and receives a binary feedback signal $y_t \in \{0, 1\}$ indicating which arm was preferred. Most work on dueling bandits is in the stochastic setting and assumes a stochastic model – a preference matrix $\mathbf{P}$ of pairwise comparison probabilities $P_{ij}$ – from which the feedback signals $y_t$ are drawn; as with standard stochastic multi-armed bandits, the target here is usually to design algorithms with $O(\ln T)$ regret bounds, and where possible, $O(\ln T)$ *anytime* (or 'horizon-free') regret bounds, for which the algorithm does not need to know the horizon or number of trials $T$ in advance.

Early work on dueling bandits often assumed strong conditions on the preference matrix $\mathbf{P}$, such as existence of a total order, under which there is a natural notion of a 'maximal' element with respect to which regret is measured. Recent work has sought to design algorithms under weaker conditions on $\mathbf{P}$; most work, however, has assumed the existence of a Condorcet winner, which is an arm $i$ that beats every other arm $j$ ($P_{ij} > \frac{1}{2} \ \forall j \neq i$), and which reduces to the maximal element when a total order exists. Unfortunately, the Condorcet winner does not always exist, and this has motivated a search for other natural notions of winners, such as Borda winners and the Copeland set (see Figure 1).[1] Among these, the only work that offers anytime $O(\ln T)$ regret bounds is the recent work of Zoghi et al. [11] on Copeland sets. In this work, we consider defining winners in dueling bandits via the natural notion of *tournament solutions* used in social choice theory, of which the Copeland set is a special case. We develop general upper confidence bound (UCB) style dueling bandit algorithms for a number of tournament solutions including the top cycle, uncovered set, and Banks set, and prove $O(\ln T)$ anytime regret bounds for them, where the regret is measured relative to the tournament solution of interest. Our proof technique is modular and can be used to develop algorithms with similar bounds for any tournament solution for which a 'selection procedure' satisfying certain 'safety conditions' can be designed. Experiments confirm the ability of our algorithms to achieve low regret relative to the target winning set of interest.

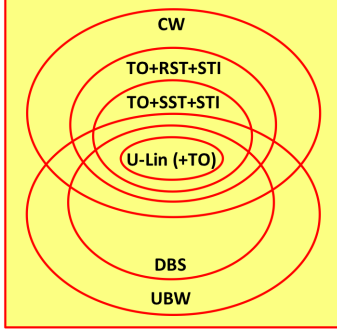

| Algorithm | Condition on P | Target Winner | Anytime? |
|---|---|---|---|
| MultiSBM [5] | U-Lin | Condorcet winner | ✓ |
| IF [1] | TO+SST+STI | Condorcet winner | ✗ |
| BTMB [2] | TO+RST+STI | Condorcet winner | ✗ |
| RUCB [6] | CW | Condorcet winner | ✓ |
| MergeRUCB [7] | CW | Condorcet winner | ✓ |
| RMED [9] | CW | Condorcet winner | ✓ |
| SECS [8] | UBW | Borda winner | ✗ |
| PBR-SE [4] | DBS | Borda winner | ✗ |
| PBR-CO [4] | Any **P** without ties | Copeland set | ✗ |
| SAVAGE-BO [3] | Any **P** without ties | Borda winner | ✗ |
| SAVAGE-CO [3] | Any **P** without ties | Copeland set | ✗ |
| CCB, SCB [11] | Any **P** without ties | Copeland set | ✓ |
| UCB-TC | Any **P** without ties | Top cycle | ✓ |
| UCB-UC | Any **P** without ties | Uncovered set | ✓ |
| UCB-BA | Any **P** without ties | Banks set | ✓ |

Figure 1: Summary of algorithms for stochastic dueling bandit problems that have $O(\ln T)$ regret bounds, together with corresponding conditions on the underlying preference matrix **P**, target winners used in defining regret, and whether the regret bounds are "anytime". The figure on the left shows relations between some of the commonly studied conditions on **P** (see Table 1 for definitions). The algorithms in the lower part of the table (shown in red) are proposed in this paper.

## 2 Dueling Bandits, Tournament Solutions, and Regret Measures

**Dueling Bandits.** We denote by $[K] = \{1, \ldots, K\}$ the set of $K$ arms. On each trial $t$, the learner selects a pair of arms $(i_t, j_t) \in [K] \times [K]$ (with $i_t$ possibly equal to $j_t$), and receives feedback in the form of a comparison outcome $y_t \in \{0, 1\}$, with $y_t = 1$ indicating $i_t$ was preferred over $j_t$ and $y_t = 0$ indicating the reverse. The goal of the learner is to select as often as possible from a set of 'good' or 'winning' arms, which we formalize below as a tournament solution.

The pairwise feedback on each trial is assumed to be generated stochastically according to a fixed but unknown pairwise preference model represented by a *preference matrix* $\mathbf{P} \in [0, 1]^{K \times K}$ with $P_{ij} + P_{ji} = 1 \, \forall i, j$: whenever arms $i$ and $j$ are compared, $i$ is preferred to $j$ with probability $P_{ij}$, and $j$ to $i$ with probability $P_{ji} = 1 - P_{ij}$. Thus for each trial $t$, we have $y_t \sim \text{Bernoulli}(P_{i_t j_t})$. We assume throughout this paper that there are no "ties" between distinct arms, i.e. that $P_{ij} \neq \frac{1}{2} \, \forall i \neq j$.[2] We denote by $\mathcal{P}_K$ the set of all such preference matrices over $K$ arms:
$$\mathcal{P}_K \;=\; \left\{ \mathbf{P} \in [0, 1]^{K \times K} : P_{ij} + P_{ji} = 1 \, \forall i, j \,;\; P_{ij} \neq \tfrac{1}{2} \, \forall i \neq j \right\}.$$
For any pair of arms $(i, j)$, we will define the margin of $(i, j)$ w.r.t. **P** as
$$\Delta_{ij}^{\mathbf{P}} = \left| P_{ij} - \tfrac{1}{2} \right|.$$

Previous work on dueling bandits has considered a variety of conditions on **P**; see Table 1 and Figure 1. Our interest here is in designing algorithms that have regret guarantees under minimal restrictions on **P**. To this end, we will consider general notions of winners that are derived from a natural tournament associated with **P**, and that are always guaranteed to exist. We will say an arm $i$ *beats* an arm $j$ w.r.t. **P** if $P_{ij} > \frac{1}{2}$; we will express this as a binary relation $\succ_{\mathbf{P}}$ on $[K]$:
$$i \succ_{\mathbf{P}} j \iff P_{ij} > \tfrac{1}{2}.$$

The *tournament associated with* **P** is then simply $\mathcal{T}_{\mathbf{P}} = ([K], E_{\mathbf{P}})$, where $E_{\mathbf{P}} = \{(i, j) : i \succ_{\mathbf{P}} j\}$. Two frequently studied notions of winners in previous work on dueling bandits, both of which are derived from the tournament $\mathcal{T}_{\mathbf{P}}$ (and which are the targets of previous anytime regret bounds), are the Condorcet winner when it exists, and the Copeland set in general:

**Definition 1** (**Condorcet winner**). *Let* $\mathbf{P} \in \mathcal{P}_K$. *If there exists an arm* $i^* \in [K]$ *such that* $i^* \succ_{\mathbf{P}} j \, \forall j \neq i^*$, *then* $i^*$ *is said to be a* Condorcet winner *w.r.t.* **P**.

**Definition 2** (**Copeland set**). *Let* $\mathbf{P} \in \mathcal{P}_K$. *The* Copeland set *w.r.t.* **P**, *denoted* $\text{CO}(\mathbf{P})$, *is defined as the set of all arms in* $[K]$ *that beat the maximal number of arms w.r.t.* **P**:
$$\text{CO}(\mathbf{P}) = \arg\max_{i \in [K]} \sum_{j \neq i} \mathbf{1}\big(i \succ_{\mathbf{P}} j\big).$$

Here we are interested in more general notions of winning sets derived from the tournament $\mathcal{T}_{\mathbf{P}}$.

Table 1: Commonly studied conditions on the preference matrix $\mathbf{P}$.

| Condition on P | Property satisfied by P |
|---|---|
| Utility-based with linear link (U-Lin) | $\exists \mathbf{u} \in [0,1]^K : P_{ij} = \frac{1-(u_i-u_j)}{2} \ \forall i,j$ |
| Total order (TO) | $\exists \sigma \in \mathcal{S}_n : P_{ij} > \frac{1}{2} \iff \sigma(i) < \sigma(j)$ |
| Strong stochastic transitivity (SST) | $P_{ij} > \frac{1}{2}, P_{jk} > \frac{1}{2} \implies P_{ik} \geq \max(P_{ij}, P_{jk})$ |
| Relaxed stochastic transitivity (RST) | $\exists \gamma \geq 1 : P_{ij} > \frac{1}{2}, P_{jk} > \frac{1}{2} \implies P_{ik} - \frac{1}{2} \geq \frac{1}{\gamma} \max(P_{ij} - \frac{1}{2}, P_{jk} - \frac{1}{2})$ |
| Stochastic triangle inequality (STI) | $P_{ij} > \frac{1}{2}, P_{jk} > \frac{1}{2} \implies P_{ik} \leq P_{ij} + P_{jk} - \frac{1}{2}$ |
| Condorcet winner (CW) | $\exists i : P_{ij} > \frac{1}{2} \ \forall j \neq i$ |
| Unique Borda winner (UBW) | $\exists i : \sum_{k \neq i} P_{ik} > \sum_{k \neq j} P_{jk} \ \forall j \neq i$ |
| Distinct Borda scores (DBS) | $\sum_{k \neq i} P_{ik} \neq \sum_{k \neq j} P_{jk} \ \forall i \neq j$ |

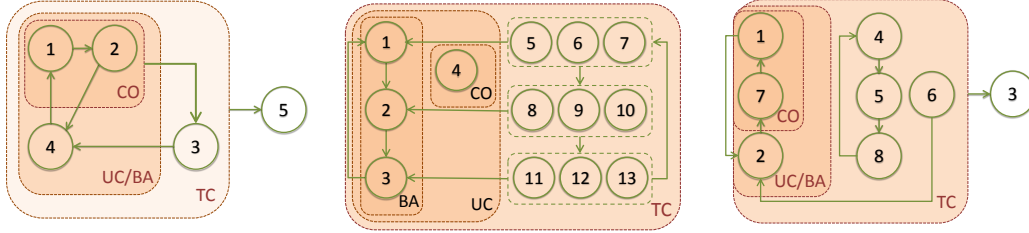

Figure 2: Examples of various tournaments together with their corresponding tournament solutions. Edges that are not explicitly shown are directed from left to right; edges that are incident on subsets of nodes (rounded rectangles) apply to all nodes within. **Left:** A tournament on 5 nodes with gradually discriminating tournament solutions. **Middle:** The Hudry tournament on 13 nodes with disjoint Copeland and Banks sets. **Right:** A tournament on 8 nodes based on ATP tennis match records.

**Tournament Solutions.** Tournament solutions have long been used in social choice and voting theory to define winners in general tournaments when no Condorcet winner exists [12, 13]. Specifically, a tournament solution is any mapping that maps each tournament on $K$ nodes to a subset of 'winning' nodes in $[K]$; for our purposes, we will define a tournament solution to be any mapping $\mathsf{S} : \mathcal{P}_K \to 2^{[K]}$ that maps each preference matrix $\mathbf{P}$ (via the induced tournament $\mathcal{T}_{\mathbf{P}}$) to a subset of winning arms $\mathsf{S}(\mathbf{P}) \subseteq [K]$.[3] The Copeland set is one such tournament solution. We consider three additional tournament solutions in this paper: the top cycle, the uncovered set, and the Banks set, all of which offer other natural generalizations of the Condorcet winner. These tournament solutions are motivated by different considerations (ranging from dominance to covering to decomposition into acyclic subtournaments) and have graded discriminative power, and can therefore be used to match the needs of different applications; see [12] for a comprehensive survey.

**Definition 3 (Top cycle).** *Let $\mathbf{P} \in \mathcal{P}_K$. The* top cycle *w.r.t. $\mathbf{P}$, denoted $\mathrm{TC}(\mathbf{P})$, is defined as the smallest set $W \subseteq [K]$ for which $i \succ_{\mathbf{P}} j \ \forall i \in W, j \notin W$.*

**Definition 4 (Uncovered set).** *Let $\mathbf{P} \in \mathcal{P}_K$. An arm $i$ is said to* cover *an arm $j$ w.r.t. $\mathbf{P}$ if $i \succ_{\mathbf{P}} j$ and $\forall k : j \succ_{\mathbf{P}} k \implies i \succ_{\mathbf{P}} k$. The* uncovered set *w.r.t. $\mathbf{P}$, denoted $\mathrm{UC}(\mathbf{P})$, is defined as the set of all arms that are not covered by any other arm w.r.t. $\mathbf{P}$:*
$$\mathrm{UC}(\mathbf{P}) = \big\{ i \in [K] : \nexists j \in [K] \text{ s.t. } j \text{ covers } i \text{ w.r.t. } \mathbf{P} \big\}.$$

**Definition 5 (Banks set).** *Let $\mathbf{P} \in \mathcal{P}_K$. A subtournament $\mathcal{T} = (V, E)$ of $\mathcal{T}_{\mathbf{P}}$, where $V \subseteq [K]$ and $E = E_{\mathbf{P}}|_{V \times V}$, is said to be* maximal acyclic *if (i) $\mathcal{T}$ is acyclic, and (ii) no other subtournament containing $\mathcal{T}$ is acyclic. Denote by $\mathrm{MAST}(\mathbf{P})$ the set of all maximal acyclic subtournaments of $\mathcal{T}_{\mathbf{P}}$, and for each $\mathcal{T} \in \mathrm{MAST}(\mathbf{P})$, denote by $m^*(\mathcal{T})$ the maximal element of $\mathcal{T}$. Then the* Banks set *w.r.t. $\mathbf{P}$, denoted $\mathrm{BA}(\mathbf{P})$, is defined as the set of maximal elements of all maximal acyclic subtournaments of $\mathcal{T}_{\mathbf{P}}$:*
$$\mathrm{BA}(\mathbf{P}) = \big\{ m^*(\mathcal{T}) : \mathcal{T} \in \mathrm{MAST}(\mathbf{P}) \big\}.$$

It is known that $\mathrm{BA}(\mathbf{P}) \subseteq \mathrm{UC}(\mathbf{P}) \subseteq \mathrm{TC}(\mathbf{P})$ and $\mathrm{CO}(\mathbf{P}) \subseteq \mathrm{UC}(\mathbf{P}) \subseteq \mathrm{TC}(\mathbf{P})$. In general, $\mathrm{BA}(\mathbf{P})$ and $\mathrm{CO}(\mathbf{P})$ may intersect, although they can also be disjoint. When $\mathbf{P}$ contains a Condorcet winner $i^*$, all four tournament solutions reduce to just the singleton set $\{i^*\}$. See Figure 2 for examples.

**Regret Measures.** When $\mathbf{P}$ admits a Condorcet winner $i^*$, the individual regret of an arm $i$ is usually defined as $r_{\mathbf{P}}^{\mathrm{CW}}(i) = \Delta_{i^*,i}^{\mathbf{P}}$, and the cumulative regret over $T$ trials of an algorithm $\mathcal{A}$ that selects arms $(i_t, j_t)$ on trial $t$ is then generally defined as $\mathcal{R}_T^{\mathrm{CW}}(\mathcal{A}) = \sum_{t=1}^{T} r_{\mathbf{P}}^{\mathrm{CW}}(i_t, j_t)$, where the pairwise regret $r_{\mathbf{P}}^{\mathrm{CW}}(i,j)$ is either the average regret $\frac{1}{2}\big(r_{\mathbf{P}}^{\mathrm{CW}}(i) + r_{\mathbf{P}}^{\mathrm{CW}}(j)\big)$, the strong regret $\max\big(r_{\mathbf{P}}^{\mathrm{CW}}(i), r_{\mathbf{P}}^{\mathrm{CW}}(j)\big)$, or the weak regret $\min\big(r_{\mathbf{P}}^{\mathrm{CW}}(i), r_{\mathbf{P}}^{\mathrm{CW}}(j)\big)$ [1, 2, 6, 7, 9].[4] When the target winner is a tournament solution $\mathsf{S}$, we can similarly define a suitable notion of individual regret of an arm $i$ w.r.t. $\mathsf{S}$, and then use this to define pairwise regrets as above.

In particular, for the three tournament solutions discussed above, we will define the following natural notions of individual regret:
$$r_{\mathbf{P}}^{\mathrm{TC}}(i) = \begin{cases} \max_{i^* \in \mathrm{TC}(\mathbf{P})} \Delta_{i^*,i}^{\mathbf{P}} & \text{if } i \notin \mathrm{TC}(\mathbf{P}) \\ 0 & \text{if } i \in \mathrm{TC}(\mathbf{P}) \end{cases}; \qquad r_{\mathbf{P}}^{\mathrm{UC}}(i) = \begin{cases} \max_{i^* \in \mathrm{UC}(\mathbf{P}): i^* \text{ covers } i} \Delta_{i^*,i}^{\mathbf{P}} & \text{if } i \notin \mathrm{UC}(\mathbf{P}) \\ 0 & \text{if } i \in \mathrm{UC}(\mathbf{P}) \end{cases};$$

$$r_{\mathbf{P}}^{\mathrm{BA}}(i) = \begin{cases} \max_{\mathcal{T} \in \mathrm{MAST}(\mathbf{P}): \mathcal{T} \text{ contains } i} \Delta_{m^*(\mathcal{T}),i}^{\mathbf{P}} & \text{if } i \notin \mathrm{BA}(\mathbf{P}) \\ 0 & \text{if } i \in \mathrm{BA}(\mathbf{P}). \end{cases}$$

In the special case when $\mathbf{P}$ admits a Condorcet winner $i^*$, the three individual regrets above all reduce to the Condorcet individual regret, $r_{\mathbf{P}}^{\mathrm{CW}}(i) = \Delta_{i^*,i}^{\mathbf{P}}$. In each case above, the cumulative regret of an algorithm $\mathcal{A}$ over $T$ trials will then be given by
$$\mathcal{R}_T^{\mathsf{S}}(\mathcal{A}) = \sum_{t=1}^{T} r_{\mathbf{P}}^{\mathsf{S}}(i_t, j_t),$$
where the pairwise regret $r_{\mathbf{P}}^{\mathsf{S}}(i,j)$ can be the average regret $\frac{1}{2}\big(r_{\mathbf{P}}^{\mathsf{S}}(i) + r_{\mathbf{P}}^{\mathsf{S}}(j)\big)$, the strong regret $\max\big(r_{\mathbf{P}}^{\mathsf{S}}(i), r_{\mathbf{P}}^{\mathsf{S}}(j)\big)$, or the weak regret $\min\big(r_{\mathbf{P}}^{\mathsf{S}}(i), r_{\mathbf{P}}^{\mathsf{S}}(j)\big)$. Our regret bounds will hold for each of these forms of pairwise regret. In fact, our regret bounds hold for any measure of pairwise regret $r_{\mathbf{P}}^{\mathsf{S}}(i,j)$ that satisfies the following three conditions:

(i) $r_{\mathbf{P}}^{\mathsf{S}}(\cdot, \cdot)$ is *normalized*: $r_{\mathbf{P}}^{\mathsf{S}}(i,j) \in [0,1]\ \forall i,j$;

(ii) $r_{\mathbf{P}}^{\mathsf{S}}(\cdot, \cdot)$ is *symmetric*: $r_{\mathbf{P}}^{\mathsf{S}}(i,j) = r_{\mathbf{P}}^{\mathsf{S}}(j,i)\ \forall i,j$; and

(iii) $r_{\mathbf{P}}^{\mathsf{S}}(\cdot, \cdot)$ is *proper w.r.t.* $\mathsf{S}$: $i,j \in \mathsf{S}(\mathbf{P}) \implies r_{\mathbf{P}}^{\mathsf{S}}(i,j) = 0$.

It is easy to verify that for the three tournament solutions above, the average, strong and weak pairwise regrets above all satisfy these conditions.[5,6]

## 3  UCB-TS: Generic Dueling Bandit Algorithm for Tournament Solutions

**Algorithm.** In Algorithm 1 we outline a generic dueling bandit algorithm, which we call UCB-TS, for identifying winners from a general tournament solution. The algorithm can be instantiated to specific tournament solutions by designing suitable selection procedures SELECTPROC-TS (more details below). The algorithm maintains a matrix $\mathbf{U}^t \in \mathbb{R}_+^{K \times K}$ of upper confidence bounds (UCBs) $U_{ij}^t$ on the unknown pairwise preference probabilities $P_{ij}$; the UCBs are constructed by adding a confidence term to the current empirical estimate of $P_{ij}$; the exploration parameter $\alpha > \frac{1}{2}$ controls the exploration rate of the algorithm via the size of the confidence terms used. On each trial $t$, the algorithm selects a pair of arms $(i_t, j_t)$ based on the current UCB matrix $\mathbf{U}^t$ using the selection procedure SELECTPROC-TS; on observing the preference feedback $y_t$, the algorithm then updates the UCBs for all pairs of arms $(i,j)$ (the UCBs of all pairs $(i,j)$ grow slowly with $t$ so that pairs that have not been selected for a while have an increasing chance of being explored).

In order to instantiate the UCB-TS algorithm to a particular tournament solution $\mathsf{S}$, the critical step is in designing the selection procedure SELECTPROC-TS in a manner that yields good regret bounds for a suitable regret measure w.r.t. $\mathsf{S}$. Below we identify general conditions on SELECTPROC-TS that allow for $O(\ln T)$ anytime regret bounds to be obtained (we will design procedures satisfying these conditions for the three tournament solutions of interest in Section 4).

**Algorithm 1** UCB-TS

---

1: **Require:** Selection procedure SELECTPROC-TS

2: **Parameter:** Exploration parameter $\alpha > \frac{1}{2}$

3: **Initialize:** $\forall\, (i,j) \in [K] \times [K]$:

$N_{ij}^1 = 0$  // # times $(i,j)$ has been compared;     $W_{ij}^1 = 0$  // # times $i$ has won against $j$;

$$U_{ij}^1 = \begin{cases} \frac{1}{2} & \text{if } i = j \\ 1 & \text{otherwise} \end{cases} \quad \text{// UCB for } P_{ij}.$$

4: **For** $t = 1, 2, \ldots$ **do:**

5:   • Select $(i_t, j_t) \leftarrow$ SELECTPROC-TS$(\mathbf{U}^t)$

6:   • Receive preference feedback $y_t \in \{0, 1\}$

7:   • Update counts:  $\forall\, (i,j) \in [K] \times [K]$:

$$N_{ij}^{t+1} = \begin{cases} N_{ij}^t + 1 & \text{if } \{i,j\} = \{i_t, j_t\} \\ N_{ij}^t & \text{otherwise} \end{cases} ; \quad W_{ij}^{t+1} = \begin{cases} W_{ij}^t + y_t & \text{if } (i,j) = (i_t, j_t) \\ W_{ij}^t + (1 - y_t) & \text{if } (i,j) = (j_t, i_t) \\ W_{ij}^t & \text{otherwise.} \end{cases}$$

8:   • Update UCBs:  $\forall\, (i,j) \in [K] \times [K]$:

$$U_{ij}^{t+1} = \begin{cases} \frac{1}{2} & \text{if } i = j \\ 1 & \text{if } i \neq j \text{ and } N_{ij}^{t+1} = 0 \\ \frac{W_{ij}^{t+1}}{N_{ij}^{t+1}} + \sqrt{\frac{\alpha \ln t}{N_{ij}^{t+1}}} & \text{otherwise.} \end{cases}$$

---

**Regret Analysis.** We show here that if the selection procedure SELECTPROC-TS satisfies two natural conditions w.r.t. a tournament solution S, namely the *safe identical-arms condition w.r.t.* S and the *safe distinct-arms condition w.r.t.* S, then the resulting instantiation of the UCB-TS algorithm has an $O(\ln T)$ regret bound for any regret measure that is normalized, symmetric, and proper w.r.t S. The first condition ensures that if the UCB matrix $\mathbf{U}$ given as input to SELECTPROC-TS in fact forms an element-wise upper bound on the true preference matrix $\mathbf{P}$ and SELECTPROC-TS returns two identical arms $(i,i)$, then $i$ must be in the winning set $\mathsf{S}(\mathbf{P})$. The second condition ensures that if $\mathbf{U}$ upper bounds $\mathbf{P}$ and SELECTPROC-TS returns two distinct arms $(i,j)$, $i \neq j$, then either both $i, j$ are in the winning set $\mathsf{S}(\mathbf{P})$, or the UCBs $U_{ij}, U_{ji}$ are still loose (and $(i,j)$ should be explored further).

**Definition 6 (Safe identical-arms condition).** *Let* $\mathsf{S} : \mathcal{P}_K \to 2^{[K]}$ *be a tournament solution. We will say a selection procedure* SELECTPROC-TS $: \mathbb{R}_+^{K \times K} \to [K] \times [K]$ *satisfies the* safe identical-arms condition w.r.t. S *if for all* $\mathbf{P} \in \mathcal{P}_K$, $\mathbf{U} \in \mathbb{R}_+^{K \times K}$ *such that* $P_{ij} \leq U_{ij}\ \forall i, j$, *we have*

$$\text{SELECTPROC-TS}(\mathbf{U}) = (i,i) \implies i \in \mathsf{S}(\mathbf{P}).$$

**Definition 7 (Safe distinct-arms condition).** *Let* $\mathsf{S} : \mathcal{P}_K \to 2^{[K]}$ *be a tournament solution. We will say a selection procedure* SELECTPROC-TS $: \mathbb{R}_+^{K \times K} \to [K] \times [K]$ *satisfies the* safe distinct-arms condition w.r.t. S *if for all* $\mathbf{P} \in \mathcal{P}_K$, $\mathbf{U} \in \mathbb{R}_+^{K \times K}$ *such that* $P_{ij} \leq U_{ij}\ \forall i, j$, *we have*

$$\text{SELECTPROC-TS}(\mathbf{U}) = (i,j),\ i \neq j \implies \left\{ (i,j) \in \mathsf{S}(\mathbf{P}) \times \mathsf{S}(\mathbf{P}) \right\} \ or \ \left\{ U_{ij} + U_{ji} \geq 1 + \Delta_{ij}^{\mathbf{P}} \right\}.$$

In what follows, for $K \in \mathbb{Z}_+$, $\alpha > \frac{1}{2}$, and $\delta \in (0, 1]$, we define

$$C(K, \alpha, \delta) = \left( \frac{(4\alpha - 1)K^2}{(2\alpha - 1)\delta} \right)^{1/(2\alpha - 1)}.$$

This quantity, which also appears in the analysis of RUCB [6], acts as an initial time period beyond which all the UCBs $U_{ij}$ upper bound $P_{ij}$ w.h.p. We have the following result (proof in Appendix A):

**Theorem 8 (Regret bound for UCB-TS algorithm).** *Let* $\mathsf{S} : \mathcal{P}_K \to 2^{[K]}$ *be a tournament solution, and suppose the selection procedure* SELECTPROC-TS *used in the UCB-TS algorithm satisfies both the safe identical-arms condition w.r.t.* S *and the safe distinct-arms condition w.r.t.* S. *Let* $\mathbf{P} \in \mathcal{P}_K$, *and let* $r_{\mathbf{P}}^{\mathsf{S}}(i,j)$ *be any normalized, symmetric, proper regret measure w.r.t.* S. *Let* $\alpha > \frac{1}{2}$ *and* $\delta \in (0, 1]$. *Then with probability at least* $1 - \delta$ *(over the feedback* $y_t$ *drawn randomly from* $\mathbf{P}$ *and any internal randomness in* SELECTPROC-TS*), the cumulative regret of the UCB-TS algorithm with exploration parameter* $\alpha$ *is upper bounded as*

$$\mathcal{R}_T^{\mathsf{S}}\big(\text{UCB-TS}(\alpha)\big) \ \leq \ C(K, \alpha, \delta) + 4\alpha\, (\ln T) \left( \sum_{i < j : (i,j) \notin \mathsf{S}(\mathbf{P}) \times \mathsf{S}(\mathbf{P})} \frac{r_{\mathbf{P}}^{\mathsf{S}}(i,j)}{(\Delta_{ij}^{\mathbf{P}})^2} \right).$$

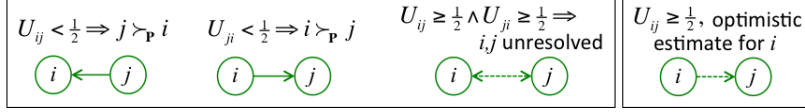

Figure 3: Inferences about the direction of preference between arms $i$ and $j$ under the true preference matrix $\mathbf{P}$ based on the UCBs $U_{ij}, U_{ji}$, assuming that $P_{ij}, P_{ji}$ are upper bounded by $U_{ij}, U_{ji}$.

## 4 Dueling Bandit Algorithms for Top Cycle, Uncovered Set, and Banks Set

Below we give selection procedures satisfying both the safe identical-arms condition and the safe distinct-arms condition above w.r.t. the top cycle, uncovered set, and Banks set, which immediately yield dueling bandit algorithms with $O(\ln T)$ regret bounds w.r.t. these tournament solutions. An instantiation of our framework to the Copeland set is also discussed in Appendix E.

The selection procedure for each tournament solution is closely related to the corresponding winner determination algorithm for that tournament solution; however, while a standard winner determination algorithm would have access to the actual tournament $\mathcal{T}_{\mathbf{P}}$, the selection procedures we design can only guess (with high confidence) the preference directions between some pairs of arms based on the UCB matrix $\mathbf{U}$. In particular, if the entries of $\mathbf{U}$ actually upper bound those of $\mathbf{P}$, then for any pair of arms $i$ and $j$, one of the following must be true (see also Figure 3):

- $U_{ij} < \frac{1}{2}$, in which case $P_{ij} \leq U_{ij} < \frac{1}{2}$ and therefore $j \succ_{\mathbf{P}} i$;
- $U_{ji} < \frac{1}{2}$, in which case $P_{ji} \leq U_{ji} < \frac{1}{2}$ and therefore $i \succ_{\mathbf{P}} j$;
- $U_{ij} \geq \frac{1}{2}$ and $U_{ji} \geq \frac{1}{2}$, in which case the direction of preference between $i$ and $j$ in $\mathcal{T}_{\mathbf{P}}$ is unresolved.

The selection procedures we design manage the exploration-exploitation tradeoff by adopting an *optimism followed by pessimism* approach, similar to that used in the design of the RUCB and CCB algorithms [6, 11]. Specifically, our selection procedures first optimistically identify a potential winning arm $a$ based on the UCBs $\mathbf{U}$ (by optimistically setting directions of any unresolved edges in $\mathcal{T}_{\mathbf{P}}$ in favor of the arm being considered; see Figure 3). Once a putative winning arm $a$ is identified, the selection procedures then pessimistically find an arm $b$ that has the greatest chance of invalidating $a$ as a winning arm, and select the pair $(a, b)$ for comparison.

### 4.1 UCB-TC: Dueling Bandit Algorithm for Top Cycle

The selection procedure SELECTPROC-TC (Algorithm 2), when instantiated in the UCB-TS template, yields the UCB-TC dueling bandit algorithm. Intuitively, SELECTPROC-TC constructs an optimistic estimate $A$ of the top cycle based on the UCBs $\mathbf{U}$ (line 2), and selects a potential winning arm $a$ from $A$ (line 3); if there is no unresolved arm against $a$ (line 5), then it returns $(a, a)$ for comparison, else it selects the best-performing unresolved opponent $b$ (line 8) and returns $(a, b)$ for comparison. We have the following result (see Appendix B for a proof):

**Theorem 9 (SELECTPROC-TC satisfies safety conditions w.r.t. TC).** SELECTPROC-TC *satisfies both the safe identical-arms condition and the safe distinct-arms condition w.r.t.* TC.

By virtue of Theorem 8, this immediately yields the following regret bound for UCB-TC:

**Corollary 10 (Regret bound for UCB-TC algorithm).** *Let* $\mathbf{P} \in \mathcal{P}_K$. *Let* $\alpha > \frac{1}{2}$ *and* $\delta \in (0, 1]$. *Then with probability at least* $1 - \delta$, *the cumulative regret of UCB-TC w.r.t. the top cycle satisfies*

$$\mathcal{R}_T^{\mathrm{TC}}\big(\mathrm{UCB\text{-}TC}(\alpha)\big) \leq C(K, \alpha, \delta) + 4\alpha\,(\ln T)\left(\sum_{i<j:(i,j)\notin \mathrm{TC}(\mathbf{P})\times \mathrm{TC}(\mathbf{P})} \frac{r_{\mathbf{P}}^{\mathrm{TC}}(i,j)}{(\Delta_{ij}^{\mathbf{P}})^2}\right).$$

### 4.2 UCB-UC: Dueling Bandit Algorithm for Uncovered Set

The selection procedure SELECTPROC-UC (Algorithm 3), when instantiated in the UCB-TS template, yields the UCB-UC dueling bandit algorithm. SELECTPROC-UC relies on the property that an uncovered arm beats every other arm either directly or via an intermediary [12]. SELECTPROC-UC optimistically identifies such a potentially uncovered arm $a$ based on the UCBs $\mathbf{U}$ (line 5); if it can be resolved that $a$ is indeed uncovered (line 7), then it returns $(a, a)$, else it selects the best-performing unresolved opponent $b$ when available (line 11), or an arbitrary opponent $b$ otherwise (line 13), and returns $(a, b)$. We have the following result (see Appendix C for a proof):

**Algorithm 2** SELECTPROC-TC

1: **Input:** UCB matrix $\mathbf{U} \in \mathbb{R}_+^{K \times K}$
2: Let $A \subseteq [K]$ be any minimal set satisfying $U_{ij} \geq \frac{1}{2} \; \forall i \in A, j \notin A$
3: Select any $a \in \arg\max_{i \in A} \min_{j \notin A} U_{ij}$
4: $B \leftarrow \{i \neq a : U_{ai} \geq \frac{1}{2} \wedge U_{ia} \geq \frac{1}{2}\}$
5: **if** $B = \emptyset$ **then**
6:     Return $(a, a)$
7: **else**
8:     Select any $b \in \arg\max_{i \in B} U_{ia}$
9:     Return $(a, b)$
10: **end if**

---

**Algorithm 3** SELECTPROC-UC

1: **Input:** UCB matrix $\mathbf{U} \in \mathbb{R}_+^{K \times K}$
2: **for** $i = 1$ **to** $K$ **do**
3:     $y(i) \leftarrow \sum_j \mathbf{1}(U_{ij} \geq \frac{1}{2}) +$
           $\sum_{j,k} \mathbf{1}(U_{ij} \geq \frac{1}{2} \wedge U_{jk} \geq \frac{1}{2})$
4: **end for**
5: Select any $a \in \arg\max_i y(i)$
6: $B \leftarrow \{i \neq a : U_{ai} \geq \frac{1}{2} \wedge U_{ia} \geq \frac{1}{2}\}$
7: **if** $\left( \forall i \neq a : (U_{ia} < \frac{1}{2}) \vee \right.$
      $\left. (\exists j : U_{ij} < \frac{1}{2} \wedge U_{ja} < \frac{1}{2}) \right)$ **then**
8:     Return $(a, a)$
9: **else**
10:     **if** $B \neq \emptyset$ **then**
11:         Select any $b \in \arg\max_{i \in B} U_{ia}$
12:     **else**
13:         Select any $b \neq a$
14:     **end if**
15:     Return $(a, b)$
16: **end if**

**Algorithm 4** SELECTPROC-BA

1: **Input:** UCB matrix $\mathbf{U} \in \mathbb{R}_+^{K \times K}$
2: Select any $j_1 \in [K]$
3: $\mathcal{J} \leftarrow \{j_1\}$    // Initialize candidate Banks trajectory
4: $s \leftarrow 1$    // Initialize size of candidate Banks trajectory
5: traj_found = FALSE
6: **while** NOT(traj_found) **do**
7:     $C \leftarrow \{i \notin \mathcal{J} : U_{ij} > \frac{1}{2} \; \forall j \in \mathcal{J}\}$
8:     **if** $C = \emptyset$ **then**
9:         traj_found = TRUE
10:         **break**
11:     **else**
12:         $j_{s+1} \in \arg\max_{i \in C}(\min_{j \in \mathcal{J}} U_{ij})$
13:         $\mathcal{J} \leftarrow \mathcal{J} \cup \{j_{s+1}\}$
14:         $s \leftarrow s + 1$
15:     **end if**
16: **end while**
17: **if** $\left( \forall 1 \leq q < r \leq s : U_{j_q, j_r} < \frac{1}{2} \right)$ **then**
18:     $a \leftarrow j_s$
19:     Return $(a, a)$
20: **else**
21:     Select any $(\widetilde{q}, \widetilde{r}) \in \arg\max_{(q,r):1 \leq q < r \leq s} U_{j_q, j_r}$
22:     $(a, b) \leftarrow (j_{\widetilde{q}}, j_{\widetilde{r}})$
23:     Return $(a, b)$
24: **end if**

**Theorem 11 (SELECTPROC-UC satisfies safety conditions w.r.t. UC).** SELECTPROC-UC *satisfies both the safe identical-arms condition and the safe distinct-arms condition w.r.t.* UC.

Again, by virtue of Theorem 8, this immediately yields the following regret bound for UCB-UC:

**Corollary 12 (Regret bound for UCB-UC algorithm).** *Let* $\mathbf{P} \in \mathcal{P}_K$. *Let* $\alpha > \frac{1}{2}$ *and* $\delta \in (0, 1]$. *Then with probability at least* $1 - \delta$, *the cumulative regret of UCB-UC w.r.t. the uncovered set satisfies*

$$\mathcal{R}_T^{\mathrm{UC}}\big(\mathrm{UCB\text{-}UC}(\alpha)\big) \;\; \leq \;\; C(K, \alpha, \delta) + 4\alpha\,(\ln T)\left( \sum_{i<j:(i,j)\notin \mathrm{UC}(\mathbf{P}) \times \mathrm{UC}(\mathbf{P})} \frac{r_{\mathbf{P}}^{\mathrm{UC}}(i,j)}{(\Delta_{ij}^{\mathbf{P}})^2} \right).$$

### 4.3 UCB-BA: Dueling Bandit Algorithm for Banks Set

The selection procedure SELECTPROC-BA (Algorithm 4), when instantiated in the UCB-TS template, yields the UCB-BA dueling bandit algorithm. Intuitively, SELECTPROC-BA first constructs an optimistic candidate maximal acyclic subtournament (set $\mathcal{J}$; also called a Banks trajectory) based on the UCBs $\mathbf{U}$ (lines 2–16). If this subtournament is completely resolved (line 17), then its maximal arm $a$ is picked and $(a, a)$ is returned; if not, an unresolved pair $(a, b)$ is returned that is most likely to fail the acyclicity/transitivity property. We have the following result (see Appendix D for a proof):

**Theorem 13 (SELECTPROC-BA satisfies safety conditions w.r.t. BA).** SELECTPROC-BA *satisfies both the safe identical-arms condition and the safe distinct-arms condition w.r.t.* BA.

Again, by virtue of Theorem 8, this immediately yields the following regret bound for UCB-BA:

**Corollary 14 (Regret bound for UCB-BA algorithm).** *Let* $\mathbf{P} \in \mathcal{P}_K$. *Let* $\alpha > \frac{1}{2}$ *and* $\delta \in (0, 1]$. *Then with probability at least* $1 - \delta$, *the cumulative regret of UCB-BA w.r.t. the Banks set satisfies*

$$\mathcal{R}_T^{\mathrm{BA}}\big(\mathrm{UCB\text{-}BA}(\alpha)\big) \;\; \leq \;\; C(K, \alpha, \delta) + 4\alpha\,(\ln T)\left( \sum_{i<j:(i,j)\notin \mathrm{BA}(\mathbf{P}) \times \mathrm{BA}(\mathbf{P})} \frac{r_{\mathbf{P}}^{\mathrm{BA}}(i,j)}{(\Delta_{ij}^{\mathbf{P}})^2} \right).$$

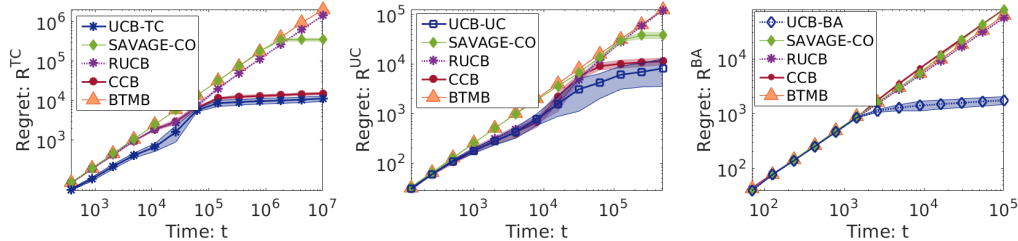

Figure 4: Regret performance of our algorithms compared to BTMB, RUCB, SAVAGE-CO, and CCB. Results are averaged over 10 independent runs; light colored bands represent one standard error. **Left:** Top cycle regret of UCB-TC on $\mathbf{P}^{\text{MSLR}}$. **Middle:** Uncovered set regret of UCB-UC on $\mathbf{P}^{\text{Tennis}}$. **Right:** Banks set regret of UCB-BA on $\mathbf{P}^{\text{Hudry}}$. See Appendix F.2 for additional results.

## 5 Experiments

Here we provide an empirical evaluation of the performance of the proposed dueling bandit algorithms. We used the following three preference matrices in our experiments, one of which is synthetic and two real-world, and none of which posesses a Condorcet winner:

- $\mathbf{P}^{\text{Hudry}} \in \mathcal{P}_{13}$: This is constructed from the Hudry tournament shown in Figure 2(b); as noted earlier, this is the smallest tournament whose Copeland set and Banks set are disjoint [14]. Details of this preference matrix can be found in Appendix F.1.1.

- $\mathbf{P}^{\text{Tennis}} \in \mathcal{P}_8$: This is constructed from real data collected from the Association of Tennis Professionals' (ATP's) website on outcomes of tennis matches played among 8 well-known professional tennis players. The tournament associated with $\mathbf{P}^{\text{Tennis}}$ is shown in Figure 2(c); further details of this preference matrix can be found in Appendix F.1.2.

- $\mathbf{P}^{\text{MSLR}} \in \mathcal{P}_{16}$: This is constructed from real data from the Microsoft Learning to Rank (MSLR) Web10K data set. Further details can be found in Appendix F.1.3.

We compared the performance of our algorithms, UCB-TC, UCB-BA, and UCB-UC, with four previous dueling bandit algorithms: BTMB [2], RUCB [6], SAVAGE-CO [3], and CCB [11].[7] In each case, we assessed the algorithms in terms of average pairwise regret relative to the target tournament solution of interest (see Section 2), averaged over 10 independent runs. A sample of the results is shown in Figure 4; as can be seen, the proposed algorithms UCB-TC, UCB-UC, and UCB-BA generally outperform existing baselines in terms of minimizing regret relative to the top cycle, the uncovered set, and the Banks set, respectively. Additional results, including results with the Copeland set variant of our algorithm, UCB-CO, can be found in Appendix F.2.

## 6 Conclusion

In this paper, we have proposed the use of general tournament solutions as sets of 'winning' arms in stochastic dueling bandit problems, with the advantage that these tournament solutions always exist and can be used to define winners according to criteria that are most relevant to a given dueling bandit setting. We have developed a UCB-style family of algorithms for such general tournament solutions, and have shown $O(\ln T)$ anytime regret bounds for the algorithm instantiated to the top cycle, uncovered set, and Banks set (as well as the Copeland set; see Appendix E).

While our approach has an appealing modular structure both algorithmically and in our proofs, an open question concerns the optimality of our regret bounds in their dependence on the number of arms $K$. For the Condorcet winner, the MergeRUCB algorithm [7] has an anytime regret bound of the form $O(K \ln T)$; for the Copeland set, the SCB algorithm [11] has an anytime regret bound of the form $O(K \ln K \ln T)$. In the worst case, our regret bounds are of the form $O(K^2 \ln T)$. Is it possible that for the top cycle, uncovered set, and Banks set, one can also show an $\Omega(K^2 \ln T)$ lower bound on the regret? Or can our regret bounds or algorithms be improved? We leave a detailed investigation of this issue to future work.

**Acknowledgments.** Thanks to the anonymous reviewers for helpful comments and suggestions. SR thanks Google for a travel grant to present this work at the conference.

## Footnotes

[1]Recently, Dudik et al. [10] also studied von Neumann winners, although they did so in a different (contextual) setting, leading to $O(T^{1/2})$ and $O(T^{2/3})$ regret bounds.

[2]The assumption of no ties was also made in deriving regret bounds w.r.t. to the Copeland set in [3, 4, 11], and exists implicitly in [1, 2] as well.

[3] Strictly speaking, the mapping $\mathsf{S}$ must be invariant under permutations of the node labels.

[4]The notion of regret used in [5] was slightly different.

[5]It is also easy to verify that defining the individual regrets as the minimum or average margin relative to all relevant arms in the tournament solution of interest (instead of the maximum margin as done above) also preserves these properties, and therefore our regret bounds hold for the resulting variants of regret as well.

[6]One can also consider defining the individual regrets simply in terms of *mistakes* relative to the target tournament solution of interest, e.g. $r_{\mathbf{P}}^{\mathrm{TC}}(i) = \mathbf{1}(i \notin \mathrm{TC}(\mathbf{P}))$, and define average/strong/weak pairwise regrets in terms of these; our bounds also apply in this case.

[7]For all the UCB-based algorithms (including our algorithms, RUCB, and CCB), we set the exploration parameter $\alpha$ to 0.51; for SAVAGE-CO, we set the confidence parameter $\delta$ to $1/T$; and for BTMB, we set $\delta$ to $1/T$ and chose $\gamma$ to satisfy the $\gamma$-relaxed stochastic transitivity property for each preference matrix.

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
