[Supplementary Material]

# Dueling Bandits: Beyond Condorcet Winners to General Tournament Solutions

## A Proof of Theorem 8

We will make use of the following two lemmas. The first lemma upper bounds the UCBs of pairs of arms that have been played sufficiently many times; the second lemma, adapted from [6], states that with high probability, after an initial transient period of play, all the UCBs upper bound the actual pairwise probabilities.

**Lemma 15.** *Let $(i, j) \in [K] \times [K]$ and $\Delta > 0$. If at any iteration $t$ of the UCB-TS algorithm, run with any selection procedure* SELECTPROC-TS, *we have $N_{ij}^t > \frac{4\alpha \ln(t)}{\Delta^2}$ , then*

$$U_{ij}^t + U_{ji}^t < 1 + \Delta\,.$$

PROOF. Let $N_{ij}^t > \frac{4\alpha \ln(t)}{\Delta^2}$. Then we have

$$
\begin{aligned}
U_{ij}^t + U_{ji}^t &= \frac{W_{ij}^t}{N_{ij}^t} + \sqrt{\frac{\alpha \ln(t)}{N_{ij}^t}} + \frac{W_{ji}^t}{N_{ji}^t} + \sqrt{\frac{\alpha \ln(t)}{N_{ji}^t}}\,, && \text{by definition of the UCBs} \\
&= 1 + 2\sqrt{\frac{\alpha \ln(t)}{N_{ij}^t}}\,, && \text{since } W_{ij}^t + W_{ji}^t = N_{ij}^t = N_{ji}^t \\
&< 1 + \Delta\,, && \text{by assumption on } N_{ij}^t.
\end{aligned}
$$

□

**Lemma 16** (Adapted from Zoghi et al., 2014 [6]). *Let $\mathbf{P} \in \mathcal{P}_K$, $\alpha > \frac{1}{2}$, and $\delta \in (0, \frac{1}{2}]$, and let $C(K, \alpha, \delta) = \left(\frac{(4\alpha-1)K^2}{(2\alpha-1)\delta}\right)^{1/(2\alpha-1)}$. Then for any selection procedure* SELECTPROC-TS, *with probability at least $1 - \delta$ (over the feedback $y_t$ drawn from $\mathbf{P}$ and any internal randomness in* SELECTPROC-TS*), the UCBs constructed by the UCB-TS algorithm satisfy*

$$\forall t \geq C(K, \alpha, \delta): \quad P_{ij} \leq U_{ij}^t \ \forall (i, j) \in [K] \times [K]\,.$$

The proof of the above lemma follows that of Zoghi et al. [6].

We are now ready to prove Theorem 8.

PROOF. [Proof of Theorem 8]
We will bound the regret conditioned on the 'good' event $\mathcal{E}$ that

$$\forall t \geq C(K, \alpha, \delta): \quad P_{ij} \leq U_{ij}^t \ \forall (i, j) \in [K] \times [K]\,;$$

the result will then follow from Lemma 16. In the following, let

$$T_0 = \lfloor C(K, \alpha, \delta) \rfloor\,.$$

As in the description of the UCB-TS algorithm, for each $\tau \in \mathbb{Z}_+$ and $(i, j) \in [K] \times [K]$, let $N_{ij}^\tau$ denote the number of trials up to trial $\tau$ in which the pair $(i, j)$ is compared:

$$N_{ij}^\tau = \sum_{t=1}^\tau \Big(\mathbf{1}\big((i_t, j_t) = (i, j)\big) + \mathbf{1}\big((i_t, j_t) = (j, i)\big)\Big)\,.$$

Then we can write the regret as

$$
\begin{aligned}
\mathcal{R}_T^{\mathsf{S}}(\text{UCB-TS}(\alpha)) &= \sum_{t=1}^{T} r_{\mathbf{P}}^{\mathsf{S}}(i_t, j_t) \\
&= \sum_{i,j} \mathbf{1}\big((i_t, j_t) = (i,j)\big) \cdot r_{\mathbf{P}}^{\mathsf{S}}(i,j) \\
&= \sum_{i \leq j} N_{ij}^{T} \, r_{\mathbf{P}}^{\mathsf{S}}(i,j), \qquad \text{since } r_{\mathbf{P}}^{\mathsf{S}}(\cdot,\cdot) \text{ is symmetric} \\
&= \sum_{i \leq j : (i,j) \notin \mathsf{S}(\mathbf{P}) \times \mathsf{S}(\mathbf{P})} N_{ij}^{T} \, r_{\mathbf{P}}^{\mathsf{S}}(i,j), \qquad \text{since } r_{\mathbf{P}}^{\mathsf{S}}(\cdot,\cdot) \text{ is proper w.r.t. } \mathsf{S} \\
&= \sum_{i \notin \mathsf{S}(\mathbf{P})} N_{ii}^{T} \, r_{\mathbf{P}}^{\mathsf{S}}(i,i) + \sum_{i < j : (i,j) \notin \mathsf{S}(\mathbf{P}) \times \mathsf{S}(\mathbf{P})} N_{ij}^{T} \, r_{\mathbf{P}}^{\mathsf{S}}(i,j). \qquad (1)
\end{aligned}
$$

We will show that, conditioned on the event $\mathcal{E}$, each of the above two terms can be bounded.

First, consider any arm $i \notin \mathsf{S}(\mathbf{P})$. Then, conditioned on $\mathcal{E}$, the safe identical-arms property of the selection procedure and Lemma 16 together ensure that after $T_0$ trials, arm $i$ is not selected for comparison with itself. Therefore, conditioned on $\mathcal{E}$, we have

$$
N_{ii}^{T} = N_{ii}^{T_0}. \qquad (2)
$$

Next, consider any pair of arms $i < j$ with $(i,j) \notin \mathsf{S}(\mathbf{P}) \times \mathsf{S}(\mathbf{P})$. In this case, conditioned on $\mathcal{E}$, the safe distinct-arms property of the selection procedure and Lemmas 15-16 together ensure that after $T_0$ trials, whenever arms $i$ and $j$ are compared on some trial $t$, we must have $N_{ij}^{t} \leq \frac{4\alpha \ln(t)}{(\Delta_{ij}^{\mathbf{P}})^2}$. Therefore, defining

$$
\begin{aligned}
A_{ij} &= \left\{ T_0 < t \leq T : \ N_{ij}^{t} \leq \tfrac{4\alpha \ln(t)}{(\Delta_{ij}^{\mathbf{P}})^2} \right\} \\
T_{ij} &= \max \left\{ t : t \in A_{ij} \right\},
\end{aligned}
$$

we have that, conditioned on $\mathcal{E}$,

$$
\begin{aligned}
N_{ij}^{T} &= N_{ij}^{T_0} + \sum_{t \in A_{ij}} \Big( \mathbf{1}\big((i_t, j_t) = (i,j)\big) + \mathbf{1}\big((i_t, j_t) = (j,i)\big) \Big) \\
&\leq N_{ij}^{T_0} + \sum_{t=1}^{T_{ij}} \Big( \mathbf{1}\big((i_t, j_t) = (i,j)\big) + \mathbf{1}\big((i_t, j_t) = (j,i)\big) \Big), \qquad \text{by definition of } T_{ij} \\
&= N_{ij}^{T_0} + N_{ij}^{T_{ij}} \\
&\leq N_{ij}^{T_0} + \frac{4\alpha \ln(T_{ij})}{(\Delta_{ij}^{\mathbf{P}})^2}, \qquad \text{since } T_{ij} \in A_{ij} \\
&\leq N_{ij}^{T_0} + \frac{4\alpha \ln(T)}{(\Delta_{ij}^{\mathbf{P}})^2}, \qquad \text{since } T_{ij} \leq T. \qquad (3)
\end{aligned}
$$

Thus, combining Eqs. (1-3), we have that, conditioned on $\mathcal{E}$,

$$
\begin{aligned}
\mathcal{R}_T^{\mathsf{S}}(\text{UCB-TS}(\alpha)) &\leq \sum_{i \notin \mathsf{S}(\mathbf{P})} N_{ii}^{T_0} \, r_{\mathbf{P}}^{\mathsf{S}}(i,i) + \sum_{i < j : (i,j) \notin \mathsf{S}(\mathbf{P}) \times \mathsf{S}(\mathbf{P})} \left( N_{ij}^{T_0} + \frac{4\alpha \ln(T)}{(\Delta_{ij}^{\mathbf{P}})^2} \right) r_{\mathbf{P}}^{\mathsf{S}}(i,j) \\
&\leq \sum_{i \leq j : (i,j) \notin \mathsf{S}(\mathbf{P}) \times \mathsf{S}(\mathbf{P})} N_{ij}^{T_0} + \sum_{i < j : (i,j) \notin \mathsf{S}(\mathbf{P}) \times \mathsf{S}(\mathbf{P})} \left( \frac{4\alpha \ln(T)}{(\Delta_{ij}^{\mathbf{P}})^2} \right) r_{\mathbf{P}}^{\mathsf{S}}(i,j), \\
&\qquad \qquad \text{since } r_{\mathbf{P}}^{\mathsf{S}}(\cdot,\cdot) \text{ is normalized} \\
&= T_0 + \sum_{i < j : (i,j) \notin \mathsf{S}(\mathbf{P}) \times \mathsf{S}(\mathbf{P})} \left( \frac{4\alpha \ln(T)}{(\Delta_{ij}^{\mathbf{P}})^2} \right) r_{\mathbf{P}}^{\mathsf{S}}(i,j).
\end{aligned}
$$

The result follows. $\qquad \square$

# B  Proof of Theorem 9

PROOF. We first prove that SELECTPROC-TC satisfies the safe identical-arms condition w.r.t. TC. Let $\mathbf{P} \in \mathcal{P}_K, \mathbf{U} \in \mathbb{R}_+^{K \times K}$ be such that $P_{ij} \leq U_{ij} \; \forall i, j$, and suppose that SELECTPROC-TC($\mathbf{U}$) = $(a, a)$. We will show that $a \in \text{TC}(\mathbf{P})$. Let the sets $A$, $B$ be defined as in SELECTPROC-TC. The identical-arms pair $(a, a)$ must be returned via line 6 of the procedure, and therefore the condition in line 5 must be satisfied, i.e. the set $B$ must be empty. Let, if possible, $a \notin \text{TC}(\mathbf{P})$ (we will show this leads to a contradiction). Then for any arm $c$, we have

$$
\begin{aligned}
c \in \text{TC}(\mathbf{P}) &\implies P_{ca} > \tfrac{1}{2}, && \text{by our assumption that } a \notin \text{TC}(\mathbf{P}) \\
&\implies U_{ca} > \tfrac{1}{2}, && \text{since } U_{ca} \geq P_{ca} \\
&\implies U_{ac} < \tfrac{1}{2}, && \text{since } B = \emptyset \text{ and therefore } c \notin B \\
&\implies c \in A, && \text{since } U_{ak} \geq \tfrac{1}{2} \; \forall k \notin A.
\end{aligned}
$$

Thus $\text{TC}(\mathbf{P}) \subseteq A$; in fact, since by our assumption, $a \in A \setminus \text{TC}(\mathbf{P})$, we have strict containment: $\text{TC}(\mathbf{P}) \subsetneq A$. Moreover, we have that for all $i \in \text{TC}(\mathbf{P}), j \notin \text{TC}(\mathbf{P}), U_{ij} \geq P_{ij} > \tfrac{1}{2}$. Thus the set $A' = \text{TC}(\mathbf{P})$ contradicts the minimality property in the definition of $A$, and therefore our assumption that $a \notin \text{TC}(\mathbf{P})$ must be false, i.e. we must have $a \in \text{TC}(\mathbf{P})$. This establishes that SELECTPROC-TC satisfies the safe identical-arms condition w.r.t. TC.

Next, we prove that SELECTPROC-TC satisfies the safe distinct-arms condition w.r.t. TC. Again, let $\mathbf{P} \in \mathcal{P}_K, \mathbf{U} \in \mathbb{R}_+^{K \times K}$ be such that $P_{ij} \leq U_{ij} \; \forall i, j$, and now suppose that SELECTPROC-TC($\mathbf{U}$) = $(a, b)$ with $a \neq b$. Let the set $B$ be defined as in SELECTPROC-TC. Now, $(a, b)$ must be returned via line 9, and therefore we must have $b \in B$. By definition of $B$, this implies that we have both $U_{ab} \geq \tfrac{1}{2}$ and $U_{ba} \geq \tfrac{1}{2}$. Thus we have

$$
\begin{aligned}
U_{ab} + U_{ba} &= \max(U_{ab}, U_{ba}) + \min(U_{ab}, U_{ba}) \\
&\geq \max(P_{ab}, P_{ba}) + \tfrac{1}{2} \\
&= (\tfrac{1}{2} + \Delta_{ab}^{\mathbf{P}}) + \tfrac{1}{2} \\
&= 1 + \Delta_{ab}^{\mathbf{P}}.
\end{aligned}
$$

This establishes that SELECTPROC-TC satisfies the safe distinct-arms condition w.r.t. TC (in fact, w.r.t. any tournament solution). $\qquad \square$

# C  Proof of Theorem 11

We will need the following lemma:

**Lemma 17.** *Let $\mathbf{U} \in \mathbb{R}_+^{K \times K}$ be such that $U_{ij} + U_{ji} \geq 1 \; \forall i, j$. Let the set $B$ be constructed from $\mathbf{U}$ as in line 6 of SELECTPROC-UC. If the condition in line 7 of SELECTPROC-UC is not satisfied, then $B \neq \emptyset$.*

PROOF. As in SELECTPROC-UC, define

$$
\begin{aligned}
y(i) &= \sum_{j=1}^{K} \mathbf{1}(U_{ij} \geq \tfrac{1}{2}) + \sum_{i=1}^{K} \sum_{k=1}^{K} \mathbf{1}(U_{ij} \geq \tfrac{1}{2} \wedge U_{jk} \geq \tfrac{1}{2}) \quad \forall i \in [K] \\
a &\in \text{argmax}_i \, y(i) \,.
\end{aligned}
$$

Suppose the condition in line 7 of SELECTPROC-UC is not satisfied, i.e. suppose $\exists i \neq a$ such that $U_{ia} \geq \tfrac{1}{2}$, and for all $j$, either $U_{ij} \geq \tfrac{1}{2}$ or $U_{ja} \geq \tfrac{1}{2}$. We will show that the set $B$, defined as

$$
B = \left\{ i \neq a : U_{ai} \geq \tfrac{1}{2} \wedge U_{ia} \geq \tfrac{1}{2} \right\},
$$

is non-empty.

Let, if possible, $B$ be empty. Then for all arms $i \neq a$, either $U_{ai} < \tfrac{1}{2}$ and $U_{ia} \geq \tfrac{1}{2}$, or $U_{ai} \geq \tfrac{1}{2}$ and $U_{ia} < \tfrac{1}{2}$ (note that since $U_{ai} + U_{ia} \geq 1$, we cannot have $U_{ai} < \tfrac{1}{2}$ and $U_{ia} < \tfrac{1}{2}$). Thus all arms $i \neq a$ are 'resolved' against $a$ (under $\mathbf{U}$), and can be partitioned into a set $C$ of arms that 'beat' $a$ (under $\mathbf{U}$), and a set $D$ of arms that 'lose' to $a$ (under $\mathbf{U}$):

$$
\begin{aligned}
C &= \left\{ i \neq a : U_{ai} < \tfrac{1}{2} \wedge U_{ia} \geq \tfrac{1}{2} \right\} \\
D &= \left\{ i \neq a : U_{ai} \geq \tfrac{1}{2} \wedge U_{ia} < \tfrac{1}{2} \right\}.
\end{aligned}
$$

Next, we claim that there is an arm $c \in C$ s.t. $U_{cd} \geq \frac{1}{2}$ $\forall d \in D$. Indeed, suppose not; then we must have $\forall c' \in C, \exists d' \in D$ s.t. $U_{c'd'} < \frac{1}{2}$. This means that $a$ beats all arms in $D$ (under $\mathbf{U}$) directly (by definition of $D$), and beats all arms in $C$ (under $\mathbf{U}$) via an intermediary in $D$. But this contradicts the assumption that the condition in line 7 is not satisfied. Therefore, there must be an arm $c \in C$ s.t. $U_{cd} \geq \frac{1}{2}$ $\forall d \in D$.

Now, consider $y(a)$ and $y(c)$:

$$
\begin{aligned}
y(a) &= \sum_{i \in [K]} \mathbf{1}(U_{ai} \geq \tfrac{1}{2}) + \sum_{j \in [K],\, k \in [K]} \mathbf{1}(U_{aj} \geq \tfrac{1}{2} \wedge U_{jk} \geq \tfrac{1}{2}) \\
&= \mathbf{1}(U_{aa} \geq \tfrac{1}{2}) + \sum_{i \in D} \mathbf{1}(U_{ai} \geq \tfrac{1}{2}) + \mathbf{1}(U_{aa} \geq \tfrac{1}{2} \wedge U_{aa} \geq \tfrac{1}{2}) + \sum_{j \in D,\, k \in C \cup D} \mathbf{1}(U_{aj} \geq \tfrac{1}{2} \wedge U_{jk} \geq \tfrac{1}{2}) \\
&= 1 + |D| + 1 + \sum_{j \in D,\, k \in C \cup D} \mathbf{1}(U_{jk} \geq \tfrac{1}{2}) \\
&= 2 + |D| + \sum_{j \in D,\, k \in C \cup D} \mathbf{1}(U_{jk} \geq \tfrac{1}{2}) \, ;
\end{aligned}
$$

$$
\begin{aligned}
y(c) &= \sum_{i \in [K]} \mathbf{1}(U_{ci} \geq \tfrac{1}{2}) + \sum_{j \in [K],\, k \in [K]} \mathbf{1}(U_{cj} \geq \tfrac{1}{2} \wedge U_{jk} \geq \tfrac{1}{2}) \\
&\geq \mathbf{1}(U_{cc} \geq \tfrac{1}{2}) + \mathbf{1}(U_{ca} \geq \tfrac{1}{2}) + \sum_{i \in D} \mathbf{1}(U_{ci} \geq \tfrac{1}{2}) + \mathbf{1}(U_{cc} \geq \tfrac{1}{2} \wedge U_{cc} \geq \tfrac{1}{2}) \\
&\quad + \mathbf{1}(U_{cc} \geq \tfrac{1}{2} \wedge U_{ca} \geq \tfrac{1}{2}) + \mathbf{1}(U_{ca} \geq \tfrac{1}{2} \wedge U_{aa} \geq \tfrac{1}{2}) + \sum_{j \in D,\, k \in C \cup D} \mathbf{1}(U_{cj} \geq \tfrac{1}{2} \wedge U_{jk} \geq \tfrac{1}{2}) \\
&= 1 + 1 + |D| + 1 + 1 + 1 + \sum_{j \in D,\, k \in C \cup D} \mathbf{1}(U_{jk} \geq \tfrac{1}{2}) \\
&= 5 + |D| + \sum_{j \in D,\, k \in C \cup D} \mathbf{1}(U_{jk} \geq \tfrac{1}{2}) \, .
\end{aligned}
$$

This gives $y(c) > y(a)$. However this contradicts the choice of $a$ in line 3. Therefore our assumption that $B$ is empty must be false, i.e. it must be the case that $B \neq \emptyset$. $\qquad \square$

We will also need the following characterization of uncovered arms, which says that an arm is uncovered if and only if it beats every other arm either directly or via an intermediary (see [12, 15]):

**Lemma 18** (Shepsle and Weingast, 1984 [15])**.** *Let* $\mathbf{P} \in \mathcal{P}_K$. *Then* $w \in \mathrm{UC}(\mathbf{P})$ *if and only if for all* $i \neq w$, *either* $w \succ_{\mathbf{P}} i$ *or* $\exists j \in [K]$ *such that* $w \succ_{\mathbf{P}} j$ *and* $j \succ_{\mathbf{P}} i$.

We are now ready for the proof of Theorem 11.

PROOF. [Proof of Theorem 11]
We first prove that SELECTPROC-UC satisfies the safe identical-arms condition w.r.t. UC. Let $\mathbf{P} \in \mathcal{P}_K, \mathbf{U} \in \mathbb{R}_+^{K \times K}$ be such that $P_{ij} \leq U_{ij}$ $\forall i, j$, and suppose that SELECTPROC-UC$(\mathbf{U}) = (a, a)$. Now, $(a, a)$ must be returned via line 8 of the procedure, and therefore the condition in line 7 must be satisfied. In particular, this condition states that for all $i \neq a$, either $U_{ia} < \frac{1}{2}$, or $\exists j \in [K]$ such that both $U_{ja} < \frac{1}{2}$ and $U_{ij} < \frac{1}{2}$. This implies that for all $i \neq a$, either $P_{ia} < \frac{1}{2}$ (i.e. $a \succ_{\mathbf{P}} i$), or $\exists j \in [K]$ such that both $P_{ja} < \frac{1}{2}$ or $P_{ij} < \frac{1}{2}$ (i.e. $a \succ_{\mathbf{P}} j$ and $j \succ_{\mathbf{P}} i$). Thus the arm $a$ beats every other arm under $\mathbf{P}$ either directly or via an intermediate arm, and therefore by Lemma 18, we have $a \in \mathrm{UC}(\mathbf{P})$. This establishes that SELECTPROC-UC satisfies the safe identical-arms condition w.r.t. UC.

Next, we prove that SELECTPROC-UC satisfies the safe distinct-arms property w.r.t. UC. Again, let $\mathbf{P} \in \mathcal{P}_K, \mathbf{U} \in \mathbb{R}_+^{K \times K}$ be such that $P_{ij} \leq U_{ij}$ $\forall i, j$, and now suppose that SELECTPROC-UC$(\mathbf{U}) = (a, b)$ with $a \neq b$. Here it must be the case that the condition in line 7 is not satisfied. Let the set $B$ be defined as in SELECTPROC-UC. Since $U_{ij} + U_{ji} \geq P_{ij} + P_{ji} = 1$ $\forall i, j$, by Lemma 17, we must have $B \neq \emptyset$, and therefore $(a, b)$ must be returned via lines 11 and 15, with $b \in B$. By definition of

$B$, this implies that we have both $U_{ab} \geq \frac{1}{2}$ and $U_{ba} \geq \frac{1}{2}$. Thus we have

$$
\begin{aligned}
U_{ab} + U_{ba} &= \max(U_{ab}, U_{ba}) + \min(U_{ab}, U_{ba}) \\
&\geq \max(P_{ab}, P_{ba}) + \frac{1}{2} \\
&= (\frac{1}{2} + \Delta_{ab}^{\mathbf{P}}) + \frac{1}{2} \\
&= 1 + \Delta_{ab}^{\mathbf{P}}.
\end{aligned}
$$

This establishes that SELECTPROC-UC satisfies the safe distinct-arms condition w.r.t. UC (in fact, w.r.t. any tournament solution). □

## D  Proof of Theorem 13

PROOF.  We first prove that SELECTPROC-BA satisfies the safe identical-arms condition w.r.t. BA. Let $\mathbf{P} \in \mathcal{P}_K, \mathbf{U} \in \mathbb{R}_+^{K \times K}$ be such that $P_{ij} \leq U_{ij} \; \forall i, j$, and suppose that SELECTPROC-BA($\mathbf{U}$) = $(a, a)$. We will show $a \in \mathrm{BA}(\mathbf{P})$. Let the set $\mathcal{J} = \{j_1, \ldots, j_s\}$ be constructed as in SELECTPROC-BA. Now, $(a, a)$ must be returned via line 19 of the procedure, which means the condition in line 17 must be true. In particular, this condition states that $U_{j_q, j_r} < \frac{1}{2} \; \forall 1 \leq q < r \leq s$, which implies that $P_{j_q, j_r} < \frac{1}{2} \; \forall 1 \leq q < r \leq s$. Thus the elements of $\mathcal{J}$ satisfy $j_s \succ_{\mathbf{P}} j_{s-1} \succ_{\mathbf{P}} \ldots \succ_{\mathbf{P}} j_1$. Moreover, there cannot be any arm $i$ that beats $j_s$ under $\mathbf{P}$, since then we would have $P_{ij} > \frac{1}{2} \; \forall j \in \mathcal{J}$ and $i$ would have been added to $\mathcal{J}$ in line 7. Therefore, the set $\mathcal{J}$ is a true Banks trajectory under $\mathbf{P}$ (forms a maximally acyclic subtournament), and $a = j_s$ is its maximal element. Thus, $a \in \mathrm{BA}(\mathbf{P})$. This establishes that SELECTPROC-BA satisfies the safe identical-arms condition w.r.t. BA.

Next, we prove that SELECTPROC-BA satisfies the safe distinct-arms property w.r.t. BA. Again, let $\mathbf{P} \in \mathcal{P}_K, \mathbf{U} \in \mathbb{R}_+^{K \times K}$ be such that $P_{ij} \leq U_{ij} \; \forall i, j$, and now suppose that SELECTPROC-BA($\mathbf{U}$) = $(a, b)$ with $a \neq b$. In this case $(a, b)$ must be returned via line 23, and therefore, by construction, we must have both $U_{ab} \geq \frac{1}{2}$ and $U_{ba} \geq \frac{1}{2}$. Thus we have

$$
\begin{aligned}
U_{ab} + U_{ba} &= \max(U_{ab}, U_{ba}) + \min(U_{ab}, U_{ba}) \\
&\geq \max(P_{ab}, P_{ba}) + \frac{1}{2} \\
&= (\frac{1}{2} + \Delta_{ab}^{\mathbf{P}}) + \frac{1}{2} \\
&= 1 + \Delta_{ab}^{\mathbf{P}}.
\end{aligned}
$$

This establishes that SELECTPROC-BA satisfies the safe distinct-arms condition w.r.t. BA (in fact, w.r.t. any tournament solution). □

## E  UCB-CO: Dueling Bandit Algorithm for Copeland Set

Before describing an instantiation of our algorithmic framework designed for the Copeland set, let us briefly consider regret measures for the Copeland set:

### E.1  Copeland Regret

There are many ways to define a regret measure that is normalized, symmetric, and proper w.r.t. the Copeland set; we consider one such natural measure below. In particular, for each arm $i \in [K]$, let $c_{\mathbf{P}}(i)$ denote the Copeland score of $i$ under $\mathbf{P}$:

$$
c_{\mathbf{P}}(i) = \sum_{j \neq i} \mathbf{1}(i \succ_{\mathbf{P}} j),
$$

Let $c_{\mathbf{P}}^*$ denote the maximal Copeland score under $\mathbf{P}$:

$$
c_{\mathbf{P}}^* = \max_i c_{\mathbf{P}}(i).
$$

Then we define the individual Copeland regret of an arm $i$ as its Copeland score deficit w.r.t. $c_{\mathbf{P}}^*$, normalized to lie in $[0, 1]$:

$$
r_{\mathbf{P}}^{\mathrm{CO}}(i) = \frac{c_{\mathbf{P}}^* - c_{\mathbf{P}}(i)}{c_{\mathbf{P}}^*} \quad \forall i \in [K].
$$

This is simply a scaled version of the Copeland regret considered by Zoghi et al. [11]. Clearly, $r_{\mathbf{P}}^{\mathrm{CO}}(i) = 0 \; \forall i \in \mathrm{CO}(\mathbf{P})$, and therefore the resulting average, weak, and strong pairwise Copeland regrets $r_{\mathbf{P}}^{\mathrm{CO}}(\cdot, \cdot)$ are all proper w.r.t. the Copeland set (see Section 2).

---
**Algorithm 5** SELECTPROC-CO
---

1: **Input:** UCB matrix $\mathbf{U} \in \mathbb{R}_+^{K \times K}$
2: **for** $i = 1$ **to** $K$ **do**
3:     $c_\mathbf{U}(i) \leftarrow \sum_{j \neq i} \mathbf{1}(U_{ij} \geq \frac{1}{2})$
4: **end for**
5: Select any $a \in \operatorname{argmax}_{i \in A} c_\mathbf{U}(i)$
6: $B \leftarrow \{i \neq a : U_{ai} \geq \frac{1}{2} \wedge U_{ia} \geq \frac{1}{2}\}$
7: **if** $B = \emptyset$ **then**
8:     Return $(a, a)$
9: **else**
10:     Select any $b \in \operatorname{argmax}_{i \in B} U_{ia}$
11:     Return $(a, b)$
12: **end if**
---

### E.2 UCB-CO Algorithm

The selection procedure SELECTPROC-CO (Algorithm 5), when instantiated in the UCB-TS template, yields the UCB-CO dueling bandit algorithm. Intuitively, SELECTPROC-CO first selects a potential Copeland winner $a$ that beats the maximal number of other arms under $\mathbf{U}$ (lines 2–5); if there is no unresolved arm against $a$ (line 7), then it returns $(a, a)$ for comparison, else it selects the best-performing unresolved opponent $b$ (line 10) and returns $(a, b)$ for comparison. This selection procedure is quite similar to the selection procedure implicitly used in the RUCB algorithm [6]; indeed, if one were to assume the existence of a Condorcet winner, then it would be natural to look for an arm $a$ that beats all other arms under $\mathbf{U}$ as RUCB does (rather than look for an arm $a$ that beats a maximal number of other arms under $\mathbf{U}$ as SELECTPROC-CO does). We have the following result:

**Theorem 19** (**SELECTPROC-CO satisfies safety conditions w.r.t. CO**). *SELECTPROC-CO satisfies both the safe identical-arms condition and the safe distinct-arms condition w.r.t.* CO.

PROOF. We first prove that SELECTPROC-CO satisfies the safe identical-arms condition w.r.t. CO. Let $\mathbf{P} \in \mathcal{P}_K, \mathbf{U} \in \mathbb{R}_+^{K \times K}$ be such that $P_{ij} \leq U_{ij} \ \forall i, j$, and suppose that SELECTPROC-CO$(\mathbf{U}) = (a, a)$. We will show that $a \in \operatorname{CO}(\mathbf{P})$. Let the set $B$ be defined as in SELECTPROC-CO. The identical-arms pair $(a, a)$ must be returned via line 8 of the procedure, and therefore the condition in line 7 must be satisfied, i.e. the set $B$ must be empty. Now, consider any true Copeland winner $i^* \in \operatorname{CO}(\mathbf{P}) \subseteq \operatorname{TC}(\mathbf{P})$. Then $U_{i^*j} \geq P_{i^*j} > \frac{1}{2} \ \forall j \notin \operatorname{TC}(\mathbf{P})$, and therefore

$$c_\mathbf{U}(i^*) \geq c_\mathbf{P}(i^*) \geq K - |\operatorname{TC}(\mathbf{P})| .$$

Since by construction $a$ maximizes $c_\mathbf{U}(\cdot)$, this implies

$$c_\mathbf{U}(a) \geq K - |\operatorname{TC}(\mathbf{P})| .$$

Now, suppose, if possible, that $a \notin \operatorname{TC}(\mathbf{P})$. Then there must be some $j \in \operatorname{TC}(\mathbf{P})$ such that $U_{aj} > \frac{1}{2}$ (as otherwise, we would have $c_\mathbf{U}(a) \leq K - |\operatorname{TC}(\mathbf{P})| - 1$). But since $j \in \operatorname{TC}(\mathbf{P})$, we must also then have $U_{ja} \geq P_{ja} > \frac{1}{2}$. Thus $j$ must be unresolved against $a$, i.e. $j \in B$. However this contradicts the fact that $B$ is empty; therefore, we must have $a \in \operatorname{TC}(\mathbf{P})$. Thus we can write

$$c_\mathbf{P}(a) \ = \ K - |\operatorname{TC}(\mathbf{P})| + \sum_{j \in \operatorname{TC}(\mathbf{P})} \mathbf{1}(P_{aj} > \tfrac{1}{2}) ;$$

$$c_\mathbf{P}(i^*) \ = \ K - |\operatorname{TC}(\mathbf{P})| + \sum_{j \in \operatorname{TC}(\mathbf{P})} \mathbf{1}(P_{i^*j} > \tfrac{1}{2}) .$$

Now, we have

$$\begin{aligned}
c_\mathbf{U}(a) \ &= \ K - |\operatorname{TC}(\mathbf{P})| + \sum_{j \in \operatorname{TC}(\mathbf{P})} \mathbf{1}(U_{aj} > \tfrac{1}{2}) \\
&= \ K - |\operatorname{TC}(\mathbf{P})| + \sum_{j \in \operatorname{TC}(\mathbf{P})} \mathbf{1}(P_{aj} > \tfrac{1}{2}) + \sum_{j \in \operatorname{TC}(\mathbf{P})} \mathbf{1}(U_{aj} > \tfrac{1}{2}, P_{ja} > \tfrac{1}{2}) \\
&= \ c_\mathbf{P}(a) + \sum_{j \in \operatorname{TC}(\mathbf{P})} \mathbf{1}(U_{aj} > \tfrac{1}{2}, P_{ja} > \tfrac{1}{2}) \\
&= \ c_\mathbf{P}(a) ,
\end{aligned}$$

since $U_{aj} > \frac{1}{2}, P_{ja} > \frac{1}{2} \implies U_{aj} > \frac{1}{2}, U_{ja} > \frac{1}{2} \implies j \in B \implies B \neq \emptyset$, which is a contradiction, and therefore we must have $\sum_{j \in \text{TC}(\mathbf{P})} \mathbf{1}(U_{aj} > \frac{1}{2}, P_{ja} > \frac{1}{2}) = 0$. Similarly,

$$
\begin{aligned}
c_{\mathbf{U}}(i^*) &= K - |\text{TC}(\mathbf{P})| + \sum_{j \in \text{TC}(\mathbf{P})} \mathbf{1}(U_{i^*j} > \tfrac{1}{2}) \, ; \\
&= K - |\text{TC}(\mathbf{P})| + \sum_{j \in \text{TC}(\mathbf{P})} \mathbf{1}(P_{i^*j} > \tfrac{1}{2}) + \sum_{j \in \text{TC}(\mathbf{P})} \mathbf{1}(U_{i^*j} > \tfrac{1}{2}, P_{ji^*} > \tfrac{1}{2}) \\
&= c_{\mathbf{P}}(i^*) + \sum_{j \in \text{TC}(\mathbf{P})} \mathbf{1}(U_{i^*j} > \tfrac{1}{2}, P_{ji^*} > \tfrac{1}{2}) \, .
\end{aligned}
$$

This gives

$$
\begin{aligned}
c_{\mathbf{P}}(a) - c_{\mathbf{P}}(i^*) &= c_{\mathbf{U}}(a) - \left( c_{\mathbf{U}}(i^*) - \sum_{j \in \text{TC}(\mathbf{P})} \mathbf{1}(U_{i^*j} > \tfrac{1}{2}, P_{ji^*} > \tfrac{1}{2}) \right) \\
&= \left( c_{\mathbf{U}}(a) - c_{\mathbf{U}}(i^*) \right) + \sum_{j \in \text{TC}(\mathbf{P})} \mathbf{1}(U_{i^*j} > \tfrac{1}{2}, P_{ji^*} > \tfrac{1}{2}) \\
&\geq 0 \, .
\end{aligned}
$$

Thus $c_{\mathbf{P}}(a) \geq c_{\mathbf{P}}(i^*)$, and therefore $a \in \text{CO}(\mathbf{P})$. This establishes that SELECTPROC-CO satisfies the safe identical-arms condition w.r.t. CO.

Next, we prove that SELECTPROC-CO satisfies the safe distinct-arms condition w.r.t. CO. Again, let $\mathbf{P} \in \mathcal{P}_K, \mathbf{U} \in \mathbb{R}_+^{K \times K}$ be such that $P_{ij} \leq U_{ij} \; \forall i, j$, and now suppose that SELECTPROC-CO$(\mathbf{U}) = (a, b)$ with $a \neq b$. Let the set $B$ be defined as in SELECTPROC-CO. Now, $(a, b)$ must be returned via line 11, and therefore we must have $b \in B$. By definition of $B$, this implies that we have both $U_{ab} \geq \frac{1}{2}$ and $U_{ba} \geq \frac{1}{2}$. Thus we have

$$
\begin{aligned}
U_{ab} + U_{ba} &= \max(U_{ab}, U_{ba}) + \min(U_{ab}, U_{ba}) \\
&\geq \max(P_{ab}, P_{ba}) + \tfrac{1}{2} \\
&= (\tfrac{1}{2} + \Delta_{ab}^{\mathbf{P}}) + \tfrac{1}{2} \\
&= 1 + \Delta_{ab}^{\mathbf{P}} \, .
\end{aligned}
$$

This establishes that SELECTPROC-CO satisfies the safe distinct-arms condition w.r.t. CO (in fact, w.r.t. any tournament solution). $\qquad \square$

By virtue of Theorem 8, this immediately yields the following regret bound for UCB-CO (as for our other algorithms, the regret bound holds for all regret measures that are normalized, symmetric, and proper w.r.t. the Copeland set; here we apply it to the pairwise Copeland regret measures resulting from the individual Copeland regret defined above):

**Corollary 20** (**Regret bound for UCB-CO algorithm**). *Let $\mathbf{P} \in \mathcal{P}_K$. Let $\alpha > \frac{1}{2}$ and $\delta \in (0, 1]$. Then with probability at least $1 - \delta$, the cumulative regret of UCB-CO w.r.t. the Copeland set satisfies*

$$
\mathcal{R}_T^{\text{CO}}\big(\text{UCB-CO}(\alpha)\big) \leq C(K, \alpha, \delta) + 4\alpha \, (\ln T) \left( \sum_{i < j : (i,j) \notin \text{CO}(\mathbf{P}) \times \text{CO}(\mathbf{P})} \frac{r_{\mathbf{P}}^{\text{CO}}(i, j)}{(\Delta_{ij}^{\mathbf{P}})^2} \right) .
$$

## F   Supplement to Section 5 (Experiments)

Below we provide details of the preference matrices used in our experiments (Section F.1), and give complete experimental results (Section F.2).

### F.1   Preference Matrices Used in Our Experiments

We used three preference matrices in our experiments: $\mathbf{P}^{\text{Hudry}}$, $\mathbf{P}^{\text{Tennis}}$, and $\mathbf{P}^{\text{MSLR}}$. These matrices are described below.

#### F.1.1   $\mathbf{P}^{\text{Hudry}}$

The Hudry tournament, shown in Figure 2(b), is a well-studied tournament on 13 nodes, and has the special property that it is the smallest tournament for which the Banks and Copeland sets are

disjoint [14]. As seen in Figure 2(b), the Hudry tournament has a Copeland set of size 1, a Banks set of size 3, an uncovered set of size 4 (containing both the Copeland set and the Banks set), and a top cycle of size 13 (i.e. containing all 13 nodes). We constructed the $\mathbf{P}^{\text{Hudry}}$ preference matrix so that its induced tournament corresponded to the Hudry tournament; the pairwise probabilities were designed to give the Copeland winner only a small margin over its dominion (arms it beats), while other arms, in particular the Banks winners, had higher margins over their dominion:

$$\mathbf{P}^{\text{Hudry}} = \begin{bmatrix}
0.5 & 0.1 & 0.1 & 0.1 & 0.6 & 0.6 & 0.6 & 0.6 & 0.6 & 0.6 & 0.6 & 0.6 & 0.6 \\
0.9 & 0.5 & 0.9 & 0.1 & 0.1 & 0.1 & 0.1 & 0.9 & 0.9 & 0.9 & 0.9 & 0.9 & 0.9 \\
0.9 & 0.1 & 0.5 & 0.9 & 0.9 & 0.9 & 0.9 & 0.1 & 0.1 & 0.1 & 0.9 & 0.9 & 0.9 \\
0.9 & 0.9 & 0.1 & 0.5 & 0.9 & 0.9 & 0.9 & 0.9 & 0.9 & 0.9 & 0.1 & 0.1 & 0.1 \\
0.4 & 0.9 & 0.1 & 0.1 & 0.5 & 0.9 & 0.9 & 0.9 & 0.9 & 0.9 & 0.1 & 0.1 & 0.1 \\
0.4 & 0.9 & 0.1 & 0.1 & 0.1 & 0.5 & 0.9 & 0.9 & 0.9 & 0.9 & 0.1 & 0.1 & 0.1 \\
0.4 & 0.9 & 0.1 & 0.1 & 0.1 & 0.1 & 0.5 & 0.9 & 0.9 & 0.9 & 0.1 & 0.1 & 0.1 \\
0.4 & 0.1 & 0.9 & 0.1 & 0.1 & 0.1 & 0.1 & 0.5 & 0.9 & 0.9 & 0.9 & 0.9 & 0.9 \\
0.4 & 0.1 & 0.9 & 0.1 & 0.1 & 0.1 & 0.1 & 0.1 & 0.5 & 0.9 & 0.9 & 0.9 & 0.9 \\
0.4 & 0.1 & 0.9 & 0.1 & 0.1 & 0.1 & 0.1 & 0.1 & 0.1 & 0.5 & 0.9 & 0.9 & 0.9 \\
0.4 & 0.1 & 0.1 & 0.9 & 0.9 & 0.9 & 0.9 & 0.1 & 0.1 & 0.1 & 0.5 & 0.9 & 0.9 \\
0.4 & 0.1 & 0.1 & 0.9 & 0.9 & 0.9 & 0.9 & 0.1 & 0.1 & 0.1 & 0.1 & 0.5 & 0.9 \\
0.4 & 0.1 & 0.1 & 0.9 & 0.9 & 0.9 & 0.9 & 0.1 & 0.1 & 0.1 & 0.1 & 0.1 & 0.5
\end{bmatrix}$$

### F.1.2   $\mathbf{P}^{\text{Tennis}}$

We constructed the $\mathbf{P}^{\text{Tennis}}$ preference matrix by compiling the all-time win-loss results of tennis matches among 8 international tennis players as recorded by the Association of Tennis Professionals (ATP).[8] In particular, we considered matches among the following 8 players:

| | |
|---|---|
| 1 | Goran Ivanisevic |
| 2 | Stefan Edberg |
| 3 | Pete Sampras |
| 4 | Boris Becker |
| 5 | Andre Agassi |
| 6 | Ivan Lendl |
| 7 | Michael Chang |
| 8 | Jim Courier |

For each pair of players $i$ and $j$, we took $P_{ij}^{\text{Tennis}}$ to be the fraction of matches between $i$ and $j$ that were won by $i$. This resulted in the following pairwise preference matrix:

$$\mathbf{P}^{\text{Tennis}} = \begin{bmatrix}
0.50 & 0.47 & 0.67 & 0.53 & 0.57 & 0.83 & 0.55 & 0.73 \\
0.53 & 0.50 & 0.57 & 0.71 & 0.67 & 0.48 & 0.43 & 0.60 \\
0.33 & 0.43 & 0.50 & 0.37 & 0.41 & 0.38 & 0.40 & 0.20 \\
0.47 & 0.29 & 0.63 & 0.50 & 0.71 & 0.52 & 0.17 & 0.14 \\
0.43 & 0.33 & 0.59 & 0.29 & 0.50 & 0.75 & 0.32 & 0.58 \\
0.17 & 0.52 & 0.62 & 0.48 & 0.25 & 0.50 & 0.29 & 0.00 \\
0.45 & 0.57 & 0.60 & 0.83 & 0.68 & 0.71 & 0.50 & 0.52 \\
0.27 & 0.40 & 0.80 & 0.86 & 0.42 & 1.00 & 0.48 & 0.50
\end{bmatrix}$$

The tournament associated with $\mathbf{P}^{\text{Tennis}}$ is shown in Figure 2(c). As can be seen, this tournament has a large top cycle of 7 players (all players except Pete Sampras). The uncovered set and Banks set here are identical, and contain 3 players (Goran Ivanisevic, Stefan Edberg, and Michael Chang); of these, only 2 players (Goran Ivanisevic and Michael Chang) are in the Copeland set.

### F.1.3   $\mathbf{P}^{\text{MSLR}}$

The Microsoft Learning to Rank (MSLR) Web10K data set contains 10,000 web-search query and document pairs, each associated with 132 features; each query-document pair is labeled with a user-assigned relevance score.[9] Following the procedure adopted by Jamieson and Nowak [16], one can treat the 132 query-document features as arms, and can consider pairwise comparisons among the features/arms in terms of how well they rank pairs of documents for a given query (judged by the user-provided relevance scores). Specifically, in order to draw a pairwise comparison between features $f_i$ and $f_j$, one would randomly sample a query $q$ and two associated documents $d$ and $d'$, and and would test whether one feature is better than the other in terms of ranking $d$ and $d'$ relative to the user-assigned scores $s(q,d), s(q,d')$: if $(f_i(q,d) - f_i(q,d'))(s(q,d) - s(q,d')) > 0$ and $(f_j(q,d) - f_j(q,d'))(s(q,d) - s(q,d')) < 0$, then feature $f_i$ wins over feature $f_j$; if $(f_i(q,d) -$

Figure 5: Tournament associated with the $\mathbf{P}^{\text{MSLR}}$ preference matrix together with its tournament solutions. As in Figure 2, edges that are not explicitly shown are directed from left to right.

$f_i(q, d'))(s(q, d) - s(q, d')) < 0$ and $(f_j(q, d) - f_j(q, d'))(s(q, d) - s(q, d')) > 0$, then feature $f_j$ wins over feature $f_i$; and otherwise, there is a tie. Thus, for any pair of features $f_i$ and $f_j$, one can estimate the associated pairwise preference probability $P_{ij}$ by sampling a few queries $q$ and associated document pairs $d, d'$ and counting the fraction of times $f_i$ wins over $f_j$ (adjusting for ties by counting half a win for each tie).

In our experiments, we used a subset of 16 features, and constructed a preference matrix $\mathbf{P}^{\text{MSLR}} \in \mathcal{P}_{16}$ by randomly sampling, for each of the $\binom{16}{2}$ pairs of features, 25 queries and document pairs as above, and counting the fraction of wins for each pair. This resulted in the following preference matrix:[10]

$$\mathbf{P}^{\text{MSLR}} = \begin{bmatrix}
0.50 & 0.58 & 0.48 & 0.52 & 0.56 & 0.64 & 0.54 & 0.56 & 0.62 & 0.54 & 0.54 & 0.66 & 0.52 & 0.58 & 0.62 & 0.56 \\
0.42 & 0.50 & 0.72 & 0.60 & 0.56 & 0.66 & 0.56 & 0.64 & 0.56 & 0.64 & 0.52 & 0.68 & 0.56 & 0.54 & 0.60 & 0.54 \\
0.52 & 0.28 & 0.50 & 0.48 & 0.68 & 0.54 & 0.52 & 0.52 & 0.72 & 0.68 & 0.58 & 0.60 & 0.52 & 0.64 & 0.64 & 0.72 \\
0.48 & 0.40 & 0.52 & 0.50 & 0.54 & 0.54 & 0.54 & 0.62 & 0.66 & 0.54 & 0.58 & 0.58 & 0.52 & 0.62 & 0.64 & 0.62 \\
0.44 & 0.44 & 0.32 & 0.46 & 0.50 & 0.58 & 0.60 & 0.48 & 0.54 & 0.54 & 0.48 & 0.60 & 0.68 & 0.56 & 0.64 & 0.52 \\
0.36 & 0.34 & 0.46 & 0.46 & 0.42 & 0.50 & 0.64 & 0.60 & 0.66 & 0.46 & 0.48 & 0.54 & 0.58 & 0.52 & 0.72 & 0.64 \\
0.46 & 0.44 & 0.48 & 0.46 & 0.40 & 0.36 & 0.50 & 0.52 & 0.66 & 0.48 & 0.42 & 0.54 & 0.62 & 0.58 & 0.52 & 0.52 \\
0.44 & 0.36 & 0.48 & 0.38 & 0.52 & 0.40 & 0.48 & 0.50 & 0.54 & 0.44 & 0.58 & 0.48 & 0.58 & 0.52 & 0.60 & 0.48 \\
0.38 & 0.44 & 0.28 & 0.34 & 0.46 & 0.34 & 0.34 & 0.46 & 0.50 & 0.42 & 0.52 & 0.48 & 0.58 & 0.52 & 0.48 & 0.52 \\
0.46 & 0.36 & 0.32 & 0.46 & 0.46 & 0.54 & 0.52 & 0.56 & 0.58 & 0.50 & 0.56 & 0.58 & 0.74 & 0.60 & 0.64 & 0.52 \\
0.46 & 0.48 & 0.42 & 0.42 & 0.52 & 0.52 & 0.58 & 0.42 & 0.48 & 0.44 & 0.50 & 0.60 & 0.64 & 0.48 & 0.56 & 0.54 \\
0.34 & 0.32 & 0.40 & 0.42 & 0.40 & 0.46 & 0.46 & 0.52 & 0.52 & 0.42 & 0.40 & 0.50 & 0.48 & 0.48 & 0.48 & 0.52 \\
0.48 & 0.44 & 0.48 & 0.48 & 0.32 & 0.42 & 0.38 & 0.42 & 0.42 & 0.26 & 0.36 & 0.52 & 0.50 & 0.54 & 0.56 & 0.48 \\
0.42 & 0.46 & 0.36 & 0.38 & 0.44 & 0.48 & 0.42 & 0.48 & 0.48 & 0.40 & 0.52 & 0.52 & 0.46 & 0.50 & 0.46 & 0.42 \\
0.38 & 0.40 & 0.36 & 0.36 & 0.36 & 0.28 & 0.48 & 0.40 & 0.52 & 0.36 & 0.44 & 0.52 & 0.44 & 0.54 & 0.50 & 0.52 \\
0.44 & 0.46 & 0.28 & 0.38 & 0.48 & 0.36 & 0.48 & 0.52 & 0.48 & 0.48 & 0.46 & 0.48 & 0.52 & 0.58 & 0.48 & 0.50
\end{bmatrix}$$

The tournament associated with $\mathbf{P}^{\text{MSLR}}$ is shown in Figure 5. As can be seen, this tournament has a small top cycle of size 4, identical Banks and uncovered sets of size 3, and a Copeland set of size 2.

### F.2 Complete Experimental Results

Complete results of our experiments on the three preference matrices above are shown in Figures 6, 7, and 8, respectively. As noted in Section 5, all algorithms were assessed on average pairwise regret relative to the target tournament solution of interest, as defined in Section 2 and Appendix E. The plots show regret performance averaged over 10 independent runs, with light colored bands representing one standard error.

As can be seen, in most cases, our dueling bandit algorithms outperform existing baselines in terms of minimizing regret relative to the tournament solutions of interest. For the Copeland regret, our UCB-CO algorithm performs similarly in practice to the CCB algorithm. The SAVAGE-CO algorithm, due to its use of the confidence parameter $\delta = 1/T$ to ensure a meaningful regret bound, tends to require a large number of trials for exploration (seen in our plots as an initial high-regret period), before turning sharply to exploitation (seen as an abrupt change to a near-zero additional regret phase).

Figure 6: Regret performance on (pairwise comparison outcomes from) $\mathbf{P}^{\text{Hudry}}$. **Top left:** Uncovered set regret. **Top right:** Banks set regret. **Bottom:** Copeland set regret. (Note: We do not consider top cycle regret for $\mathbf{P}^{\text{Hudry}}$ since in this case the top cycle is the entire set of arms.)

Figure 7: Regret performance on (pairwise comparison outcomes from) $\mathbf{P}^{\text{Tennis}}$. **Top left:** Top cycle regret. **Top right:** Uncovered set regret. **Bottom left:** Banks set regret. **Bottom right:** Copeland set regret.

Figure 8: Regret performance on (pairwise comparison outcomes from) $\mathbf{P}^{\mathrm{MSLR}}$. **Top left:** Top cycle regret. **Top right:** Uncovered set regret. **Bottom left:** Banks set regret. **Bottom right:** Copeland set regret.

## Footnotes

[8] ATP website: http://www.atpworldtour.com

[9] This data set is available from: http://research.microsoft.com/en-us/projects/mslr/

[10]Occasionally, preference matrices generated by sampling from real-world data may contain "tied" preferences, i.e. pairs $i \neq j$ with empirically observed $P_{ij} = P_{ji} = \frac{1}{2}$. In such cases, we suggest breaking ties by adding small random perturbations to $P_{ij}$ and $P_{ji}$ using the minimal margin between non-tied arms, defining $P'_{ij} = \frac{1}{2} + \epsilon\Delta_{\min}$ and $P'_{ji} = \frac{1}{2} - \epsilon\Delta_{\min}$, where $\Delta_{\min} = \min_{(i,j):P_{ij}\neq\frac{1}{2}} \Delta_{ij}$ and $\epsilon$ is a Rademacher random variable taking values in $\{\pm 1\}$ with equal probability. This results in a tie-free perturbed matrix $\mathbf{P}'$ which retains the same minimal margin as the original tied matrix $\mathbf{P}$.