[Reviews · NeurIPS 2016]

Reviewer 1

Summary

The paper generalizes dueling bandit approaches from current solution concepts (condorcet winners) to general tournament solutions. A detailed overview over possible tournament solutions is provided, which are shown to be adressable by UCB-style algorithms.

Qualitative Assessment

The paper is remarkably well written. It provides detailed background on dueling bandits, existing solution concepts, typical assumptions, and the general tournament solutions that this paper extends to. In addition to the obtained algorithms and bounds, the authors present a generic recipe for solutions that address a range of tournament solutions. Minor comments: - The specific tournament solutions addressed here could be motivated in more detail, i.e., in what practical settings would each be preferred? This would make it easier to judge potential impact of the paper. - footnote 1 mentions that [10] applies in a different setting - a bit more detail would be helpful to allow readers to understand the difference between these settings. - The proposed algorithms make use of the idea of "optimism followed by pessimism". This ideas was the basis of existing approaches (RUCB) which should be cited when this concept is explained in section 4 to clarify that this is an existing concept.

Confidence in this Review

2-Confident (read it all; understood it all reasonably well)


Reviewer 2

Summary

This paper deals with the dueling bandit problem in situations where the goal is to find other notions of winners than the Condorcet one (which is not guaranteed to exist) or the Copeland one (which does always exist, but has certain flaws). Unlike the MAB problem, the question of what constitutes the best arm(s) is not completely straightforward to answer in the dueling setting, in particular when there is no single arm that is preferred to all other arms (aka the Condorcet winner). Indeed, the field of social choice theory has devised a near infinite supply of such notions of winners: the reason for this overabundance of definitions is that in the absence of a Condorcet winner, it is difficult to find a definition that satisfies all reasonable properties that one might hope for, so various definitions have been proposed that satisfy various sets of desirable properties. Given that, it is natural to seek algorithms that can find these other winners, since depending on the application domain, it might very well be that the Condorcet winner does not exist and that the Copeland winner is undesirable. With this in mind, the authors propose a flexible algorithm that is capable of accommodating such needs, given certain rather weak assumptions on the sought after notion of winner. In plain terms, the algorithm can find solution concepts that are stable with respect to optimistic estimates of the underlying preference matrix. This result can be seen as a refinement of the ideas in reference [3] in the paper. The main difference is that while the SAVAGE algorithm in [3] was restricted to the finite horizon setting, the algorithm in this paper does not need knowledge of the horizon of the experiment in advance. Moreover, the authors conduct experiments on three examples comparing their algorithm adapted to three specific notions of winners against three existing algorithms, although the description of the experiments is lacking in some important details. First of all, it is unclear whether the comparison is being carried out against the Condorcet or the Copeland version of SAVAGE, and if the former is the case, why was there no comparison against Copeland SAVAGE? Secondly, I do not understand how RUCB could have sublinear regret on the Sushi and MSLR examples, since it crucially assumes the existence of a Condorcet winner, which is absent in those examples. Finally, it is rather disappointing that the authors do not compare against the state of the art duleing bandit algorithm(s) in the Condorcet case, proposed in reference [9] in the paper.

Qualitative Assessment

Overall, the theoretical/algorithmic portions of the paper are written reasonably clearly and the arguments seem correct to me. On the issue of novelty, the domain into which the paper is venturing is relatively novel, but the proofs techniques are not extremely original, although there is definitely value in using existing ideas to solve a new problem. There are also some issues with the experiments (as I mentioned above), so I urge the authors to revisit that portion of the paper.

Confidence in this Review

3-Expert (read the paper in detail, know the area, quite certain of my opinion)


Reviewer 3

Summary

This paper proposes a dueling bandits setting with new solution concepts that generalize beyond Condorcet winners and Copeland winners. These include top cycle winners, uncovered set winners, and Banks set winners. The paper introduces a generic UCB-style learning algorithm that can be instantiated for each of the various solution concepts. Regret bounds are given, as are strong empirical results.

Qualitative Assessment

This is a strong paper, with an elegant solution to an elegant problem formulation. I have a few questions/comments for the authors. -- The regret bounds seem to scale as K^2 log T, which seems unavoidable in such general settings. Are there ways to refine the proof and/or subroutines so that the regret scales as K log T instead, when there are more assumptions? -- Can the authors comment on the connection between the proposed approach and RUCB? Superficially, the structure of the two algorithms seem related. -- I imagine another baseline is action elimination via pure exploration, i.e., randomly explore pairs of viable arms (those that have not yet been eliminated from being part of the solution). One reason I'm suggesting this is because such a strategy might also have a regret bound that scales as K^2 log T. -- There is room in the paper to discuss the implications of the theoretical guarantees, such as the quadratic scaling w.r.t. K or what the C constant is. *** I thank the authors for answering my questions in their response ***

Confidence in this Review

3-Expert (read the paper in detail, know the area, quite certain of my opinion)


Reviewer 4

Summary

The key contribution of this article is to propose and analyze a generic and anytime (i.e. "horizon-free") algorithm called "Upper Confidence Bound for Tournament Solution" (UCB-TS) for the K-armed voting/dueling bandits with preference cycles. This meta algorithm maintains a confidence bound matrix on the pairwise win/loss probabilities and delegates the choice of the duels to a specialized sub-function called SelectProc-TS. If this selection procedure satisfies two conditions, namely the "Safe-identical arm condition" (Definition 6) and the "Safe distinct arms condition" (Definition 7), the anytime regret of UCB-TS is upper bounded by O(K^2 \log{T}) (Theorem 8). The authors decline UCB-TS on three "Tournament Solutions" (or preference elucidation systems) by providing specific selection procedures: Top Cycle, Uncovered Set and Banks set. Some simulations on ranking datasets follow.

Qualitative Assessment

The definition of regret as |\Delta_{i^*,i}-1/2| is natural when i^* is a Condorcet winner but \Delta_{i^*,i}-1/2 may become negative in other situations like in Figure 2 (a) where the arm 4 is winning against arm 1 which is the minimal element of the Copleand winning set. A simpler notion of regret would be to use directly the indicator function of the non-winning set. I liked the modular structure of the proofs: 1 - define a meta-algorithm UCB-TS and give minimal conditions to control its regret; 2 - show that these conditions are fulfilled by the considered tournament solutions. - Obtaining K^2 log(T) regret bound is however not a surprising result, the main difficulty with dueling bandits is to obtain a K log(T) regret bound like it was done for RUCB. - Is UCB-Copeland equivalent to RUCB when applied on a Condorcet bandit? - Is there a generic methodology to build the selection procedure from a tournament solution? On Figure 1 page 2, the authors made a real effort to provide a complete bibliography, but they apparently misunderstood the fact that SAVAGE is also a generic algorithm which works with any tournament solution including Copeland, Top Cycle, Uncovered set and Banks set. It is true however that the generic sampling sensitivity criterion may be difficult to code. If the paper is accepted, the author should precise that they only used the Copeland version of SAVAGE in their experiments.

Confidence in this Review

3-Expert (read the paper in detail, know the area, quite certain of my opinion)


Reviewer 5

Summary

The authors proposed a family of UCB-style dueling bandits algorithms for general tournament solutions. These tournament solutions include Copeland set as a special case and always exist. They also proved the any time regret bound for these algorithms is O(log T).

Qualitative Assessment

Overall, this paper is well written and this work is of practical interest to the community. Line 208 a typo: should BTMB be "BTM"? The numerical results for both Sushi and MSLR dataset are weak. The proposed algorithms only slightly outperform RUCB. It would be better if the authors could add error bars to the plot. Generally Copeland Confidence Bound(CCB) outperforms RUCB in identifying a Copeland winner. It will strengthen the paper if the authors can include CCB as a benchmark. One other thing is that all the experiments considered in this paper have less than 20 arms. Are the proposed algorithms scalable for a large number of arms? It is interesting to see how they perform compared with SCB (Scalable Copeland Bandits) when the number of arms is large. Update: I have read the author's feedback and my review doesn't change.

Confidence in this Review

2-Confident (read it all; understood it all reasonably well)


Reviewer 6

Summary

The authors consider dueling bandits under three different tournament solutions in social choice theory, in particular, beyond the well-studied concordet and copeland winners. They consider a class of UCB style algorithms for the dueling bandit problem in these settings and prove near-optimal regret bounds.

Qualitative Assessment

*Thanks to the authors for their response -- I have edited my review below accordingly, demarked with stars.* The paper is very well written, with a thorough literature review that takes the reader through a somewhat complex landscape, and proofs that are easy to follow both at a high- and low-level. Overall, the push towards wider notions of "winners" from social choice theory, and the minimal assumptions on the preference matrix P make the result appealing. Indeed this approach seems promising, potentially for other tournament solutions. Some discussion on why these particular solutions were chosen, and whether the authors indeed believe similar results could be attained for other settings would be interesting. *Please do add a discussion on this in the final version.* Overall the bounds are not surprising -- the main difficulties seem to be overcome similarly to the RUCB approach, and the novelty is in showing that the necessary conditions are fulfilled by the given tournament solutions. Am I missing something here? *I read the rebuttal on this point, and still feel that the main difficulties were overcome in the RUCB proof -- from my perspective, the contribution here is the "modularization" of the proof in a way that it can be applied to other settings.* The experimental results were a nice addition, especially the selection of the data sets for the preference matrices. However, I was not sure why each target was only considered for one preference matrix and/or in which combination -- I would prefer the 3x3 comparison or some justification why not all were evaluated. Moreover, as RUCB and the new algorithms are relatively close in performance, it would be good to present error bars. Lastly, I value the push to beyond Concordet/Copeland, but still it would be good to use CCB as a benchmark as it sometimes outperforms RUCB. *Please do make these changes in the final version.*

Confidence in this Review

2-Confident (read it all; understood it all reasonably well)